# Binarized Convolutional Neural Networks with Channel Quadrupling and Smooth Downsampling

## Abstract

This paper proposes novel binarized convolutional neural networks (BCNNs) named **QB-Net** and **QSB-Net**, specifically designed to **Q**uadruple the number of channels and incorporate a so-called **S**mooth downsampling in **B**CNNs for low-cost mobile environments. The proposed models combine FP32 depthwise separable (DS) convolutions with binarized $1 \times 1$ pointwise convolutions, offering reduced computational costs in the pointwise convolutions. To enhance the degraded performance of the above naive combination, the proposed models start with a small number of channels in shallow layers and expand them during downsampling by a factor of four, effectively managing model complexity in the downsampling. The proposed model structure maintains low computational costs in the shallow blocks and increases model complexity in the deep blocks, providing a wider dynamic range to manage information in the frequency domain. As a result, the proposed models overcome the limitations of existing BCNNs, delivering improved performance while reducing the total computational costs. For further performance enhancements, we propose a novel smooth downsampling with heightwise and widthwise sequential downsampling steps, doubling the number of channels at each step. Besides, we show that the channelwise self-attention (SE) is applicable with minimal additional computational costs in the proposed models. Besides, multiple binarized convolutions in the fully-connected (FC) layer reduce storage costs without requiring 8-bit quantized convolutions. Experimental results demonstrate the efficiency of the proposed models in terms of performance, computational costs, and inference latency on real hardware. Notably, the QSB-Net-Large with SE achieve 71.2% Top-1 accuracy on ImageNet-1K and 69.2 mean intersection over union (mIoU) in the semantic segmentation on the PASCAL VOC dataset, outperforming other counterparts.

## 1 Introduction

Although substantial parallelism in GPUs or specialized accelerators can achieve a considerable speedup, edge devices lack sufficient parallelism for accommodating the increasing model complexity. BCNNs binarize both weights and activations into $-1$ and $+1$ in binarized convolutions. As a result, multiply-accumulate operations are replaced by bitwise XNOR and bit-counting operations, leveraging bit-level parallelism in CPU-based devices. While earlier BCNNs experienced significant accuracy drops compared with their FP32 counterparts, recent BCNNs achieve good performance comparable to mobile-friendly CNNs. However, as computational costs increase within BCNNs, the expected inference speedup could not be satisfactory. For example, whereas ReActNetA (Liu et al., 2020) adopts binarized $3 \times 3$ spatial convolutions, its FP32 counterpart, MobileNetV1 (Howard et al., 2017) utilizes FP32 DS spatial convolutions. Compared with FP32 DS convolutions, binarized $3 \times 3$ spatial convolutions require more computations, failing to provide faster inference on real hardware. Therefore, novel lightweight structures should be studied to minimize computational costs while avoiding significant performance degradation in BCNNs.

Conventional CNNs typically double the number of channels or adopt customized channel expansion during downsampling (He et al., 2016; Ridnik et al., 2021) to mitigate information loss. In BCNNs, however, we expect that the binarization error and low resolution of BCNNs require a dif-

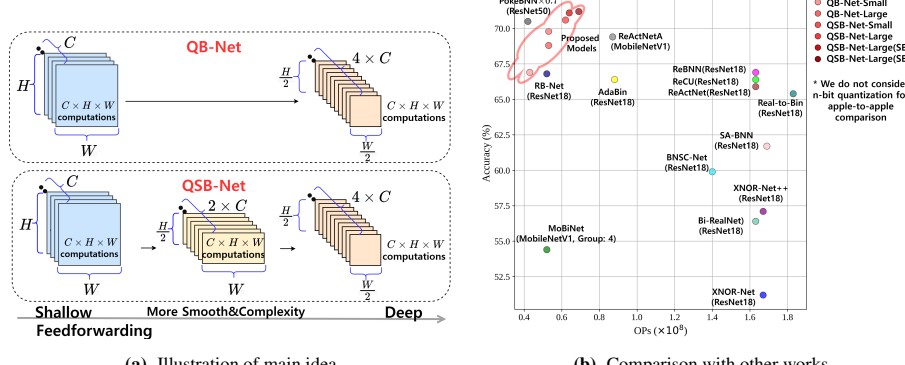

**(a).** Illustration of main idea        **(b).** Comparison with other works

**Figure 1:** Main idea and comparison in terms of Top-1 accuracy and OPs on ImageNet-1K (Russakovsky et al., 2015). In Figure 1 (b), the model in parentheses is the FP32 counterpart of its binarized model.

ferent strategy for performing the channel expansion and downsampling. Our hypothesis suggests that the performance of BCNNs can be significantly influenced by the complexity of deep blocks and the configuration of downsampling. Although increasing the number of channels of all blocks by the same ratio can mitigate the issue, computational costs dramatically increase. Therefore, we think that developing a novel strategy for channel expansion and downsampling has the potential to achieve significant benefits in BCNNs. The approach in the conceptual illustration of Figure 1 (a) quadruply increases the number of channels in deeper blocks during downsampling. By having a small number of channels in the shallow block, the proposed strategy can reduce the total computations compared with existing BCNNs.

This paper proposes novel BCNNs named **QB-Net** and **QSB-Net**, quadrupling the number of channels in channel expansion and configuring smooth downsampling in the new structures of BCNNs. The newly developed designs and main contributions are as follows:

1. **New Structure for Channel Quadrupling:** To reduce the computations in pointwise convolutions, the proposed models utilize $3 \times 3$ DS convolutions and binarized pointwise convolutions. However, the existing channel expansion with the above structure shows significant performance degradation. To overcome the problem, the proposed models have a small number of channels in the shallow blocks and increase the number of channels by a factor of four during downsampling. The small number of channels in the shallow blocks reduces the total computations, as shown in Figure 1 (b). The channel quadrupling can provide a wider dynamic range in deep blocks, enhancing the ability to manage the information in the frequency domain and achieving good performance in BCNNs.

2. **Smooth Downsampling and Techniques for Better Models:** In QSB-Net, the proposed smooth downsampling processes two sequential steps involving heightwise and widthwise 1-D downsampling, producing noticeable performance enhancements. Each 1-D downsampling layer can double the number of channels, resulting in quadrupled channels after the two steps of downsampling in Figure 1 (a). Experimental results show that channelwise self-attention in DS convolutions is effective, as shown in Figure 1 (b). Besides, 8-bit quantization can be valid for the FP32 DS convolutional and FC layers. Notably, the usage of multiple binarized convolutions in the FC layer can reduce storage costs, which can solve the issue of the increasing number of channels in the FC layer.

3. **Experiments on Various Datasets and Real Hardware:** We evaluated the proposed models for image classification and semantic segmentation. The experiments on a Raspberry Pi 4B (RPi 4B) and a Samsung Exynos processor using Larq Compute Engine (LCE) (Bannink et al., 2021) show that the proposed models have significant inference speedup compared with the baseline model and mobile-friendly CNNs.

The proposed model reaches up to 71.2% Top-1 accuracy on ImageNet-1K. and 69.2 mIoU in semantic segmentation on the PASCAL VOC dataset, outperforming other counterparts. The above results prove that the proposed channel quadrupling and smooth downsampling can be effective in BCNNs.

## 2 RELATED WORKS

Courbariaux et al. (2016) showed that BCNNs can reduce memory usage and energy consumption by nearly $32\times$ compared with baselined FP32 models. Many existing works have focused on improving the performance of BCNNs due to their significant performance drops. XNOR-Net (Rastegari et al., 2016) adopted a scaling factor for both weights and input features of binarized convolutional layers, achieving 51.4% Top-1 accuracy with binarized ResNet18 (He et al., 2016) on ImageNet-1K. Bi-RealNet (Liu et al., 2018) utilized the single skipped shortcut, enhancing the Top-1 accuracy of binarized ResNet18 up to 56.4%. XNOR-Net++ (Bulat et al., 2019) advanced BCNNs by introducing heightwise and widthwise scaling factors for the output features. Real-to-Bin (Martinez et al., 2019) enhanced the performance up to 65.4% by incorporating a self-attention block.

Whereas the above BCNNs are based on ResNet (He et al., 2016), MobiNet (Phan et al., 2020a) and ReActNetA (Liu et al., 2020) followed the structure of MobileNetV1 (Howard et al., 2017). However, MobiNet produced only 53.5% Top-1 accuracy on ImageNet-1K. ReActNetA (Liu et al., 2020) achieved 69.4% Top-1 accuracy by deploying binarized $3 \times 3$ convolutions instead of using FP32 DS convolutions. The outstanding performance of Real-to-Bin and ReActNetA showed the effectiveness of teacher-student training (Hinton et al., 2015) in BCNNs. There have been several works to develop specific activation functions and training methods for BCNNs. IR-Net (Qin et al., 2020) introduced a binarization method and so-called error decaying estimator for the training. SI-BNN (Wang et al., 2020) proposed a binarization function with trainable thresholds. BNSC-Net (Wu et al., 2021) decomposed 2-D convolutions to employ additional skip connections. RB-Net (Liu et al., 2022) reshaped pointwise convolutions and incorporated a balanced activation function. ReCU (Xu et al., 2021) employed a rectified clamp unit to improve its training results. SA-BNN (Liu et al., 2021) mitigated the weight flip issues in BCNNs. AdaBin (Tu et al., 2022) adaptively optimized weights and features to follow the value distributions of its FP32 baseline model. ReBNN (Xu et al., 2023) introduced weighted reconstruction loss to reduce the weight oscillation during training. Whereas PokeBNN (Zhang et al., 2022) achieved impressive results by adopting ResNet50 as its baseline and teacher models, its hyperparameter for the clipping bound and stride configuration in the first convolutional layer introduced a different optimization strategy. Different from Zhang et al. (2022), the proposed QSB-Net shows a novel strategy for channel quadrupling and smooth downsampling. Therefore, we conclude that the fundamental techniques and contributions of the proposed models are totally different from those of PokeBNN.

## 3 BACKGROUNDS

For a binarized convolution in BCNNs, both filter weights and input features are quantized into binary values. For given binarized input features $I_b$ and binarized $K \times K$ filters $F_b$, the conventional binarized convolution, denoted as $I_b * F_b$, can be formulated as:

$$(I_b * F_b)(i, j) = \gamma \cdot \sum_{c=0}^{C-1} \sum_{m=0}^{K-1} \sum_{n=0}^{K-1} I_b(c, i+m, j+n) \cdot F_b(c, m, n), \tag{1}$$

where $i, j$ are the spatial indices of the output feature map in a channel. The term $\gamma$ is the scaling factor of the binarized convolutional outputs. During inference, whereas pre-binarized weights $F_b$ can be pre-stored, input features $I$ are binarized into $I_b$ during feedforwarding. While so-called binarization-aware training updates real-valued weights in the backpropagation during training, the updated real-valued weights are binarized during feedforwording. Let $\epsilon(a, b)$ be the error between $a$ and $b$. Binarization errors of $\epsilon(F(c, m, n), F_b(c, m, n))$ exist between the real-valued updated weights and the binarized weights used during feedforwarding.

Several works studied better value distributions for the binarized convolution (Helwegen et al., 2019; Liu et al., 2021; Tu et al., 2022; Xu et al., 2023; Zhang et al., 2022). However, the calibration of value distributions requires specific hyperparameters, so we think that there could be no consistently valid rules. A customized sign function (Liu et al., 2018; 2020) requires a long model training time due to the complex derivatives of the customized sign function. We do not focus on new methods for calibrating value distributions and customized activation functions. Without using the above new methods and hyperparameters, we adopt a straight-through estimator (STE) (Bengio et al., 2013) to approximate the derivatives of the sign conversion. Besides, we assume that the output features of the binarized convolution are scaled with channelwise learnable scaling factors.

In order to show the computational complexity, the number of floating-point operations (FLOPs) has been generally used. In BCNNs, when BOPs denote the number of binarized multiply-accumulate operations, $OPs = FLOPs + \frac{BOPs}{64}$ are used to estimate the total computations. However, the latencies on real hardware platforms were not reduced by a factor of $\frac{1}{64}$ (Bannink et al., 2021). On the other hand, entropy is commonly used to quantify the average amount of information produced by a stochastic data source. Although the entropy could not be directly related to the model performance, we believe that it could demonstrate the information characteristics in the proposed models. To explain the information characteristics of the proposed model, the entropy of the output features in each block is analyzed in Appendix A.3.

In BCNNs, increasing model complexity has been proven to produce better model performance, suppressing negative effects of the binarization error (Lin et al., 2017). In other words, when the number of values to be summed increases in the binarized convolution in Eq.(1), the output representation capacity can be enhanced, offsetting the binarization errors. The enhanced representation capacity can capture different aspects of features that exist in a wide range of the frequency domain. Although the information loss due to the low resolution of the binarized convolutions can be mitigated by capturing the features in the high-frequency domain, additional computational costs are required. By increasing the depth multiplier (Howard et al., 2017), the number of channels in all blocks is multiplied, which can enhance the performance of BCNNs. However, the total computations quadruply increase, which can be a burden in mobile-friendly BCNNs.

## 4 QSB-NET WITH CHANNEL QUADRUPLING & SMOOTH DOWNSAMPLING

### 4.1 MOTIVATIONS

The depth multiplier for multiplying the number of channels in all blocks dramatically increases the total computations, having long inference latency in mobile-friendly BCNNs. Whereas most existing BCNNs have focused on training techniques and specific blocks on the baseline model structure, the efficient BCNN structures were not sufficiently studied. To reduce the total computations, the structural development of BCNNs should mainly increase the model complexity of the efficient part that significantly contributes to model performance. On the other hand, we hypothesize that information loss during downsampling can impact model performance. Although channel expansion during downsampling can improve representation capacity, it is questionable whether the doubled or customized channel expansions in conventional CNNs are the most effective in BCNNs. Besides, the binarization error could compound the effects of information loss during downsampling. Therefore, we develop a novel strategy for configuring channel expansion and downsampling in BCNNs.

### 4.2 BLOCKS FOR CHANNEL QUADRUPLING AND SMOOTH DOWNSAMPLING

Depending on the configurations of the number of channels and smooth downsampling, we develop four models named QB-Net-Small, QB-Net-Large, QSB-Net-Small, and QSB-Net-Large. Figure 2 illustrates the block structures used in QB-Net. The structure of MobileNetV1 (Howard et al., 2017) inspires the usage of DS convolutions in BCNNs. MobileNetV1 (Howard et al., 2017) deploys both DS convolutions and $1 \times 1$ pointwise convolutions. Besides, DS convolutions account for only 3% of the total computations in MobileNetV1. Several existing BCNNs were motivated by the structure of MobileNet (Phan et al., 2020a;b; Liu et al., 2020). However, these works adopted binarized $3 \times 3$ convolution (Liu et al., 2020) or grouped binarized convolution (Phan et al., 2020a;a) instead of using FP32 DS convolution. In Liu et al. (2020), as the number of channels increased, the binarized $3 \times 3$ convolutions dramatically increase the total computations compared with the usage of DS convolutions. In Phan et al. (2020a;b), the grouped binarized DS convolutions did not achieve acceptable accuracy compared with MobileNetV1.

Considering the small computational portion of the DS convolutions in MobileNetV1, we do not prioritize the binarization of DS convolutions. Therefore, we determine that the proposed models deploy FP32 $3 \times 3$ DS convolutional layer (**DSCONV**) in a block. However, a naive modification of the baseline model can show significant performance degradation. When applying FP32 DS convolutions instead of binarized $3 \times 3$ convolutions in ReActNetA (Liu et al., 2020), the modification of ReActNetA only achieved 64.2% Top-1 accuracy on ImageNet-1K using teacher-student train-

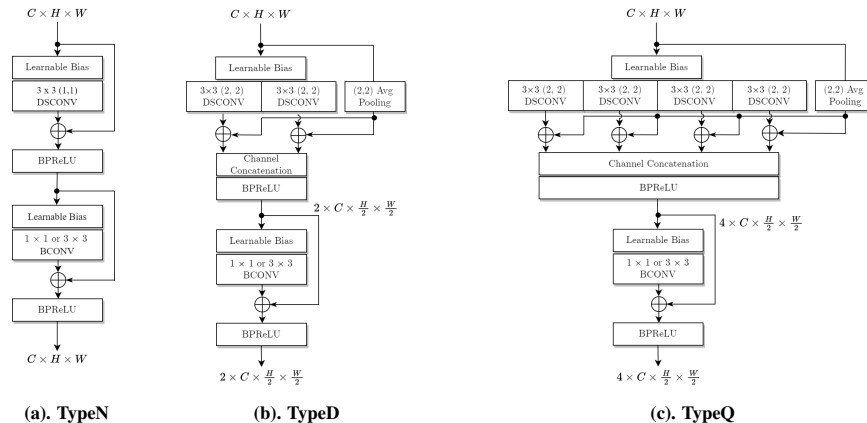

(a). TypeN  (b). TypeD  (c). TypeQ

**Figure 2:** Illustration of blocks used in QS-Net. Terms **DSCONV** and **BCONV** refer to the DS convolutional and binarized convolutional layers, respectively. Suffixes **N**, **D**, and **Q** mean **normal**, **double in downsampling**, and **quadruple in downsampling**, respectively. While block **TypeN** does not change the shape of output features, blocks **TypeD** and **TypeQ** expand channels along with downsampling feature maps. When the number of input channels $C$ is small (e.g., 16 or 32), $3 \times 3$ BCONV is used instead of $1 \times 1$ BCONV. The values in parentheses are the heightwise and widthwise strides, respectively. Terms $H$ and $W$ denote the height and width of feature maps. During downsampling, the shortcut connection utilizes the 2-D average pooling layer with a stride of $(2, 2)$. The learnable bias layer (Liu et al., 2020) is deployed just before its convolutional layer. BPReLU layers are used after the shortcut is added.

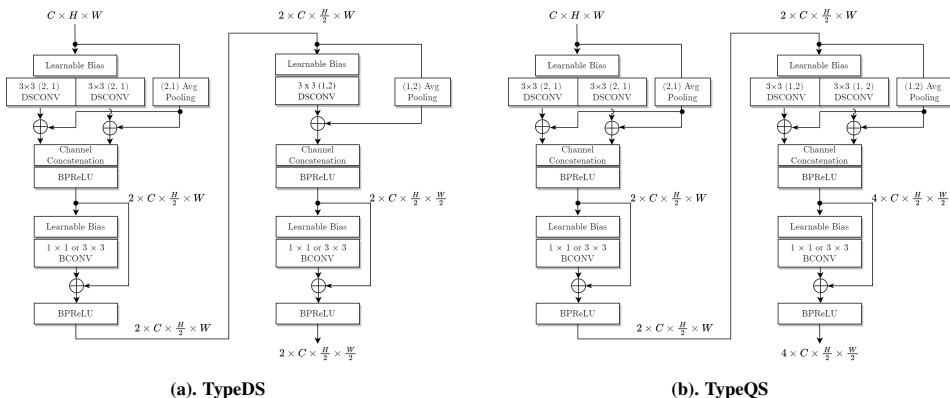

(a). TypeDS  (b). TypeQS

**Figure 3:** Illustration of blocks used in QSB-Net. Suffixes **DS** and **QS** denote **double in smooth downsampling** and **quadruple in smooth downsampling**, respectively. For the smooth downsampling, DS convolutions are sequentially performed with strides of $(2, 1)$ and $(1, 2)$. In the second DSCONV, whereas the number of channels is doubled in **TypeQS**, **TypeDS** does not expand the channels.

ing (Hinton et al., 2015). Therefore, a novel structural breakthrough is required to achieve better performance while maintaining small total computational costs.

Figure 2 illustrates **TypeN**, **TypeD**, and **TypeQ** blocks deployed in QB-Net. In the TypeD and TypeQ blocks, after performing downsampling with a stride of $(2, 2)$ in DSCONVs, the output channels from DSCONVs are concatenated to expand the number of channels. After performing the convolutions, batch normalization (BN) is performed in both DSCONV and BCONV, which is not shown in Figure 2 for clarity.

While QB-Net deploys TypeN, TypeD, and TypeQ blocks, the proposed QSB-Net can utilize other blocks named **TypeDS** and **TypeQS** during downsampling. Figure 3 illustrates the block structures in QSB-Net, which support so-called smooth downsampling. Whereas QB-Net performs conventional downsampling with a stride of $(2, 2)$, QSB-Net adopts two-stage downsampling to provide additional model complexity during downsampling. In TypeQS, two DSCONVs are performed sequentially with strides of $(2, 1)$ and $(1, 2)$. A BCONV is deployed after performing each DSCONV. In a block, after performing the first DSCONV and channel concatenation, the shape of feature maps can be $2 \times C \times \frac{H}{2} \times W$. Then, the downsampling in the second DSCONV produces the feature

**Table 1:** Comparison of model structures. QB-Net-Small and QSB-Net-Small quadruple the number of channels in TypeQ and TypeQS blocks. In the Input and Output columns, each term denotes $C \times H \times W$ in Figure 2.

| Index | ReActNet, MobileNetV1 | | | QB-Net-Small[1] | | | QSB-Net-Small[1] | | |
|---|---|---|---|---|---|---|---|---|---|
| | Block | Input | Output | Block | Input | Output | Block | Input | Output |
| 1 | Conv[2] | $3 \times 224 \times 224$ | $32 \times 112 \times 112$ | Conv[2] | $3 \times 224 \times 224$ | $16 \times 112 \times 112$ | Conv[2] | $3 \times 224 \times 224$ | $16 \times 112 \times 112$ |
| 2 | Reduction[3] | $32 \times 112 \times 112$ | $64 \times 112 \times 112$ | TypeN | $16 \times 112 \times 112$ | $16 \times 112 \times 112$ | TypeN | $16 \times 112 \times 112$ | $16 \times 112 \times 112$ |
| 3 | Reduction | $64 \times 112 \times 112$ | $128 \times 56 \times 56$ | TypeD | $16 \times 112 \times 112$ | $32 \times 56 \times 56$ | TypeDS | $16 \times 112 \times 112$ | $32 \times 56 \times 56$ |
| 4 | Normal | $128 \times 56 \times 56$ | $128 \times 56 \times 56$ | TypeN | $32 \times 56 \times 56$ | $32 \times 56 \times 56$ | TypeN | $32 \times 56 \times 56$ | $32 \times 56 \times 56$ |
| 5 | Reduction | $128 \times 56 \times 56$ | $256 \times 28 \times 28$ | TypeQ | $32 \times 56 \times 56$ | $128 \times 28 \times 28$ | TypeQS | $32 \times 56 \times 56$ | $128 \times 28 \times 28$ |
| 6 | Normal | $256 \times 28 \times 28$ | $256 \times 28 \times 28$ | TypeN | $128 \times 28 \times 28$ | $128 \times 28 \times 28$ | TypeN | $128 \times 28 \times 28$ | $128 \times 28 \times 28$ |
| 7 | Reduction | $256 \times 28 \times 28$ | $512 \times 14 \times 14$ | TypeQ | $128 \times 28 \times 28$ | $512 \times 14 \times 14$ | TypeQS | $128 \times 28 \times 28$ | $512 \times 14 \times 14$ |
| 8 | Normal | $512 \times 14 \times 14$ | $512 \times 14 \times 14$ | TypeN | $512 \times 14 \times 14$ | $512 \times 14 \times 14$ | TypeN | $512 \times 14 \times 14$ | $512 \times 14 \times 14$ |
| 9 | Normal | $512 \times 14 \times 14$ | $512 \times 14 \times 14$ | TypeN | $512 \times 14 \times 14$ | $512 \times 14 \times 14$ | TypeN | $512 \times 14 \times 14$ | $512 \times 14 \times 14$ |
| 10 | Normal | $512 \times 14 \times 14$ | $512 \times 14 \times 14$ | TypeN | $512 \times 14 \times 14$ | $512 \times 14 \times 14$ | TypeN | $512 \times 14 \times 14$ | $512 \times 14 \times 14$ |
| 11 | Normal | $512 \times 14 \times 14$ | $512 \times 14 \times 14$ | TypeN | $512 \times 14 \times 14$ | $512 \times 14 \times 14$ | TypeN | $512 \times 14 \times 14$ | $512 \times 14 \times 14$ |
| 12 | Normal | $512 \times 14 \times 14$ | $512 \times 14 \times 14$ | TypeN | $512 \times 14 \times 14$ | $512 \times 14 \times 14$ | TypeN | $512 \times 14 \times 14$ | $512 \times 14 \times 14$ |
| 13 | Reduction | $512 \times 14 \times 14$ | $1024 \times 7 \times 7$ | TypeQ | $512 \times 14 \times 14$ | $2048 \times 7 \times 7$ | TypeQS | $512 \times 14 \times 14$ | $2048 \times 7 \times 7$ |
| 14 | Normal | $1024 \times 7 \times 7$ | $1024 \times 7 \times 7$ | TypeN | $2048 \times 7 \times 7$ | $2048 \times 7 \times 7$ | TypeN | $2048 \times 7 \times 7$ | $2048 \times 7 \times 7$ |
| 15 | AvgPool | $1024 \times 7 \times 7$ | $1024 \times 1 \times 1$ | AvgPool | $2048 \times 7 \times 7$ | $2048 \times 1 \times 1$ | AvgPool | $2048 \times 7 \times 7$ | $2048 \times 1 \times 1$ |
| 16 | FC | $1024 \times 1 \times 1$ | 1000 | FC | $2048 \times 1 \times 1$ | 1000 | FC | $2048 \times 1 \times 1$ | 1000 |

[1] Whereas QB-Net-Small and QSB-Net-Small adopt TypeQ and TypeQS in the 13-th block, QB-Net-Large and QSB-Net-Large deploy them in the 9-th block and set the number of channels as 2048 from the 10-th block.

[2] The first block denoted as **Conv** consists of the conventional FP32 convolution, batch normalization, and ReLU layers.

[3] There is a channel expansion without downsampling in the first Reduction block.

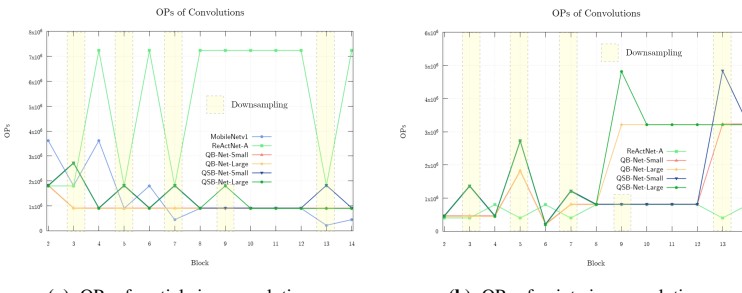

**(a).** OPs of spatialwise convolutions     **(b).** OPs of pointwise convolutions

**Figure 4:** OPs of convolutions in each block. Yellow regions indicate the blocks that perform downsampling. Whereas Figure 4 (a) shows the OPs of the DS convolution in the proposed models, Figure 4 (b) visualizes the OPs of the next binarized pointwise convolution. In Figure 4, the proposed QB-Net-Small, QB-Net-Large, QSB-Net-Small, and QSB-Net-Large do not decrease OPs in the downsampling blocks. Other counterparts such as MobileNetV1 (Howard et al., 2017) and ReActNetA (Liu et al., 2020) have small OPs in the downsampling blocks. It is noted that only QB-Net-Large and QSB-Net-Large perform downsampling in the 9-th block.

maps of $4 \times C \times \frac{H}{2} \times \frac{W}{2}$ after performing the second channel concatenation. On the other hand, in the second DSCONV of TypeDS, the number of channels is not doubled.

Several layers in the blocks are based on previous works as follows: the single skipped shortcut (Liu et al., 2018) is used for each DSCONV and BCONV, where symbol $\oplus$ denotes the elementwise addition. A learnable bias (Liu et al., 2020) is used to calibrate the distribution of input features in each channel. The activation layer with PReLU-BN, denoted as **BPReLU**, is deployed after the addition with a shortcut. In Phan et al. (2020a), the activation layer with PReLU-BN is used. However, its effects were not compared with other works such as RPReLU (Liu et al., 2020). In our experiments, BPReLU slightly outperformed RPReLU by 0.1%-0.3% in the image classification on ImageNet-1K.

## 4.3 MODEL STRUCTURE WITH CHANNEL QUADRUPLING

In order to achieve better performance, the number of channels can be expanded to increase the representative capacity. However, when the numbers of channels in all blocks are naively multiplied, its computational costs dramatically increase. Therefore, the model structure is important to determine which block has more channels, considering both computational costs and model performance. In existing FP32 models, the channel expansion during downsampling is either doubled (He et al., 2016) or customized (Sandler et al., 2018; Huang et al., 2019; Tan & Le, 2019). However, many existing BCNNs are based on the structure of ResNet (He et al., 2016) or MobileNetV1 (Howard et al., 2017), so previous studies about new structures seem to be insufficient. Table 1 shows the model structures of the proposed QS-Net-Small and QSB-Net-Small, comparing with MobileNetV1

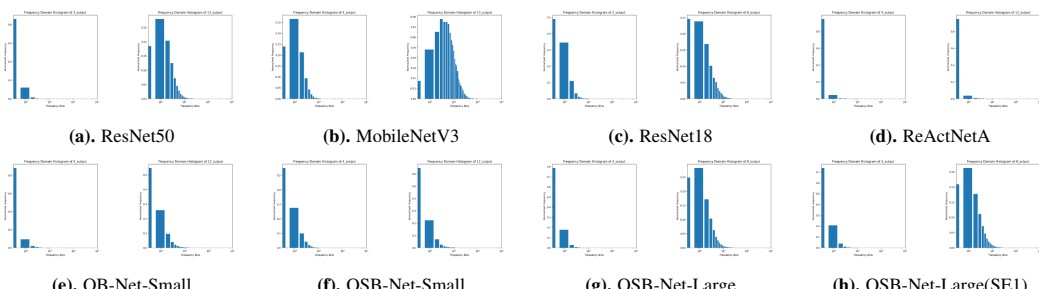

**Figure 5:** Frequency domain histograms of proposed models and counterparts in the outputs of the third (left) and final (right) downsampling blocks. The output features of a goose image are hooked and used to obtain the frequency domain histograms. In FP32 models, wide dynamic ranges are shown in the output of the final downsampling blocks. Whereas ReActNetA (Liu et al., 2020) shows a narrow range in the frequency domain in the output of the final downsampling block, the proposed models have wide dynamic ranges. The proposed Large models have wider dynamic ranges than the Small models in the frequency domain.

and ReActNet (Liu et al., 2020). Because QB-Net-Small and QSB-Net-Small adopt TypeQ and TypeQS in the 13-th block, TypeQ and TypeQS blocks have $2048 \times 7 \times 7$ output feature maps. Compared with MobileNetV1 and ReActNet, when $H$ and $W$ of feature maps are large, the number of channels $C$ is small. On the other hand, when $H$ and $W$ of feature maps are small, $C$ increases by quadrupling channels during downsampling. QB-Net-Large and QSB-Net-Large adopt TypeQ and TypeQS in the 9-th block and set the number of output channels as 2048. The model structures of QB-Net-Large and QSB-Net-Large are summarized in Appendix A.1.

## 4.4 INCREASED REPRESENTATION CAPACITY IN DEEP BLOCKS

Figure 4 illustrates the OPs of the convolutions in each block, representing the main idea of the QB-Net and QSB-Net. Whereas the counterparts such as MobileNetV1 (Howard et al., 2017) and ReActNetA (Liu et al., 2020) reduce OPs in the downsampling blocks, the channel expansion in QB-Net does not decrease OPs in the downsampling block. Moreover, additional computations are performed in the TypeDS and TypeQS blocks. We expect that the representation capacity in a block of BCNNs during downsampling can have a significant impact on the model performance. The low resolution of binarized convolutions could degrade representation capacity, so it is thought that the low resolution can be compensated with additional model complexity in the proposed blocks. The proposed model structures increase model complexity for deep blocks during downsampling. Several existing works show that increasing representation capacity during downsampling is helpful for achieving better performance in BCNNs. For example, XNOR-Net (Rastegari et al., 2016) and Bi-RealNet (Liu et al., 2018) deploy FP32 $1 \times 1$ convolutional layer for the shortcut connection. However, the shortcut connection with FP32 $1 \times 1$ convolutions significantly increases the total OPs, showing long latencies on real mobile hardware (Bannink et al., 2021). We note that the channel quadrupling can increase the representation capacity during downsampling without using the FP32 $1 \times 1$ convolutions. It is expected that the proposed structure using the channel quadrupling can have a wider dynamic range in the frequency domain from the increased complexity in deep blocks. Although the number of channels in the deep blocks increases, the total computations are significantly reduced by having a small number of channels in shallow blocks.

To show the effects of the increased representation capacity and wider dynamic range in deep blocks, the frequency domain histograms are compared. Figure 5 shows the frequency domain histograms of proposed models and counterparts in the outputs of the third (left) and final (right) downsampling blocks. Notably, the proposed models have wider dynamic ranges than ReActNetA (Liu et al., 2020). The proposed QSB-Net-Large and QSB-Net-Large(SE1) have dynamic ranges comparable to those of FP32 models. However, the similarity between Figures 5 (g) and (h) indicates that the self-attention block does not widen the dynamic range in the frequency domain. Therefore, we conclude that the role of the channel quadrupling is critical to obtain wide dynamic ranges in the frequency domain. Besides, the feature maps of deep blocks are visualized in Appendix A.8.

**Table 2:** Comparison with existing BCNNs on ImageNet-1K. The names of the baseline FP32 models are in parentheses.

| Model | Top-1 (%) | FLOPs ($\times10^8$) | BOPs ($\times10^8$) | OPs ($\times10^8$) | Model | Top-1 (%) | FLOPs ($\times10^8$) | BOPs ($\times10^8$) | OPs ($\times10^8$) |
|---|---|---|---|---|---|---|---|---|---|
| ResNet18 | 69.6 | 18.2 | - | 18.2 | MobileNetV1 | 70.6 | 5.75 | - | 5.75 |
| MobileNetV2 | 71.8 | 3.0 | - | 3.0 | SuffleNetV2 | 69.4 | 1.46 | - | 1.46 |
| GhostNet | 73.9 | 1.41 | - | 1.41 | AlphaNet-A0 | 77.8 | 2.03 | - | 2.03 |
| XNOR-Net(ResNet18) | 51.2 | 1.41 | 17.0 | 1.67 | MobiNet-Mid(MobileNetV1)[1] | 54.4 | 0.52 | - | 0.52 |
| Bi-RealNet(ResNet18) | 56.4 | 1.54 | 16.8 | 1.63 | SI-BNN(Bi-RealNet) | 59.7 | 1.54 | 16.8 | 1.63 |
| XNOR-Net++(ResNet18) | 57.1 | 1.41 | 17.0 | 1.67 | BNSC-Net(ResNet18) | 59.9 | - | - | 1.40 |
| RB-Net(ResNet18) | 66.8 | - | - | 0.52 | RB-Net(ResNet34) | 70.2 | - | - | 0.71 |
| IR-Net(ResNet18) | 58.1 | - | - | 1.63 | SA-BNN(ResNet18) | 61.7 | - | - | 1.69 |
| Real-to-Bin(ResNet18) | 65.4 | 1.56 | 1.68 | 1.83 | AdaBin(ResNet18) | 66.4 | - | - | 0.88 |
| ReBNN(ResNet18) | 66.9 | - | - | 1.63 | ReActNetA(MobileNetV1)[1] | 69.4 | 0.12 | 48.2 | 0.87 |
| PokeBNN$\times0.75$(ResNet50)[2] | 70.5 | 0.10 | 20.3 | 0.42 | ReCU(ResNet18) | 66.4 | - | - | 1.63 |
| **QB-NET-Small** | **66.9** | **0.28** | **9.3** | **0.43** | **QSB-NET-Small** | **68.8** | **0.36** | **10.8** | **0.53** |
| **QB-NET-Large** | **69.8** | **0.29** | **15.5** | **0.53** | **QSB-NET-Large** | **70.6** | **0.36** | **16.9** | **0.62** |
| **QSB-NET-Large(SE1)[3]** | **71.1** | **0.37** | **16.9** | **0.64** | **QSB-NET-Large(SE2)[3]** | **71.2** | **0.42** | **16.9** | **0.69** |

[1] Instead of DS convolutions in MobileNetV1 (Howard et al., 2017), MobiNet-Mid (Phan et al., 2020a) and ReActNetA (Liu et al., 2020) adopted grouped convolutions and binarized $3 \times 3$ convolutions, respectively.

[2] Both the baseline and teacher models are ResNet50 (He et al., 2016). In PokeBNN (Zhang et al., 2022), the first convolution adopted a stride of 4, which made PokeBNN have small FLOPs. We did not consider $n$-bit quantized operations in this comparison, counting $n$-bit quantized operations as FLOPs.

[3] While term **SE2** denotes that self-attention (SE) blocks are deployed after all DS convolutions, term **SE1** indicates that SE blocks are deployed only after the DS convolutions during downsampling.

## 4.5 APPLICABILITY OF TECHNIQUES FOR BETTER MODELS

The applicability of the techniques for achieving enhanced performance can be considered with small additional computations. Whereas DSCONVs during downsampling reduce the computational costs by removing channelwise operations, existing BCNNs (Phan et al., 2020a; Liu et al., 2020) adopted the channelwise spatial convolutions. However, the channelwise operations significantly increase computational costs, so it could be better to deploy the self-attention (SE) block (Hu et al., 2018) after the DSCONVs, applying the attention mechanism to different channels with small computational costs. Because the number of channels is 2048 in the FC layer, its parameters are doubled compared with MobileNetV1 (Howard et al., 2017) and ReActNetA (Liu et al., 2020). Any fixed-point format in the FC layer can reduce storage costs, so we performed an experiment to know whether the FC layer with 8-bit quantized weights is applicable, which is explained in Ablations Studies. Besides, the FC layer with 8-bit quantized weights can be implemented with 8 binarized convolutions, which is described in Appendix A.2. The usage of binarized convolutions in the FC layer reduces storage costs of the FC layer by $\times4$ without using 8-bit fixed-point operations.

## 5 EXPERIMENTAL RESULTS AND ANALYSIS

### 5.1 IMAGE CLASSIFICATION AND LATENCY ON REAL HARDWARE

We experimented with the proposed models on ImageNet-1K. Like ReActNetA (Liu et al., 2020), experiments adopted the two-stage teacher-student training using pretrained ResNet34 as a teacher (Hinton et al., 2015). In the first stage with 256 epochs, whereas the input features for BCONVs were binarized, weights were FP32 values. In the second stage, the pretrained weights from the first stage were used in the initialization. Both input features and weights for BCONVs were binarized during 256 training epochs. The detailed training process is described in Appendix A.4.

Table 2 shows the comparison with several mobile-friendly CNNs and BCNNs on ImageNet-1K, where a dash (-) denotes that the value was not reported in the reference. The FP32 models and their accuracies are listed above the first midline. The proposed models can significantly reduce OPs compared with ShuffleNetV2 (Ma et al., 2018) and GhostNet (Han et al., 2020). Mostly, the proposed models outperformed existing BCNNs above the second midline in terms of OPs. Although MobiNet-Mid (Phan et al., 2020a) had small OPs, Top-1 accuracy was only 54.4%. ReActNetA (Liu et al., 2020) needed additional OPs due to the usage of binarized $3 \times 3$ convolutions instead of DS convolutions. PokeBNN$\times0.75$ (Zhang et al., 2022) only had 0.42 $\times10^8$ OPs. Unlike the proposed

**Table 3:** Comparison of parameters and latency for ImageNet-1K images.

| Model | Parameters (Mbyets) | OPs ($\times 10^8$) | Latency(ms) RPi 4B | Latency(ms) Exynos | Top-1 (%) |
|---|---|---|---|---|---|
| MobileNetV1 | 16.9 | 5.75 | 160.8 | 34 | 70.6 |
| MobileNetV2 | 14.0 | 3.0 | 117.4 | 23 | 71.8 |
| XNOR-Net(ResNet18) | 4.2 | 1.67 | 87.0 | 21 | 51.2 |
| Bi-RealNet(ResNet18) | 4.2 | 1.63 | 80.2 | 19 | 56.4 |
| Real-to-Bin(ResNet18) | 5.4 | 1.83 | 100.9 | - | 65.4 |
| ReActNetA | 7.7 | 0.87 | 120.4 | 30 | 69.4 |
| QuickNet-Small | 4.0 | - | 17.5 | 9 | 59.4 |
| QuickNet | 4.2 | - | 27.4 | 13 | 63.3 |
| **QB-Net-Small** | **9.8(3.5)** [2] | **0.43** | **55.5** | **14** | **66.9** |
| **QB-Net-Large** | **12.0(5.8)** [2] | **0.53** | **65.5** | **14** | **69.8** |
| **QSB-Net-Small** | **10.0(3.9)** [2] | **0.53** | **76.1** | **18** | **68.8** |
| **QSB-Net-Large** | **12.3(6.2)** [2] | **0.62** | **86.2** | **21** | **70.6** |
| **QSB-Net-Large(SE1)** | **12.4(6.3)** [2] | **0.69** | **89.7** | **22** | **71.2** |

[1] In Bannink et al. (2021), QuickNet-Small and QuickNet had 59.4% and 63.3% Top-1 accuracies on ImageNet-1K.

[2] The value in parentheses denotes the storage costs when 8-bit quantization was applied to the FC layer. 8 binarized convolutions can replace the 8-bit quantized FC layer. In the ablation studies, the quantization showed no significant accuracy drop.

**Table 4:** Comparison of semantic segmentations on PASCAL VOC dataset.

| Model | W/F [1] | mIoU | Model | W/F [1] | mIoU |
|---|---|---|---|---|---|
| ResNet18 | 32/32 | 64.9 | LQ-Net | 3/3 | 62.5 |
| GroupNet | 1/1 | 60.5 | GroupNet+BPAC [2] | 1/1 | 65.1 |
| CBNN(Sum) [3] | 1/1 | 66.2 | CBNN(Cat) [3] | 1/1 | 66.5 |
| ReActNetA | 1/1 | 61.8 | | | |
| **QB-Net-Small** | **1/1** | **62.3** | **QSB-Net-Small** | **1/1** | **66.2** |
| **QB-Net-Large** | **1/1** | **68.1** | **QSB-Net-Large** | **1/1** | **69.2** |

[1] Terms W and F indicate $n$-bit quantization of weights and features.

[2] Term BPAC represents binary parallel atrous convolution.

[3] Terms Sum and Cat denote the summation and concatenation

and other existing models, PokeBNN adopted a stride of 4 in the first FP32 convolutional layer, which was the main reason for having the small OPs.

Table 3 summarizes the comparison in terms of parameters and latency using Larq Compute Engine (LCE) (Bannink et al., 2021) on a single thread of RPi 4B and a Samsung Exynos-9820 processor. We note that the supported layers in LCE were limited, so only several mobile-friendly CNNs and BCNNs based on ResNet18 and MobileNetV1 were compared in Table 3. All proposed models were faster than FP32 models in Table 3. QB-Net showed good efficiency in terms of Top-1 accuracy and latency. For example, QB-Net-Large can achieve 69.8% Top-1 accuracy on ImageNet-1K, having $0.53 \times 10^8$ OPs and 65.5 ms latency on the RPi 4B. Although QSB-Net-Large can enhance Top-1 accuracy by 0.8%, its latency increased by 20.7 ms. Because QSB-Net-Large(SE1) only adopted SE blocks during downsampling, the increasing latency was small. Compared with Re-ActNetA (Liu et al., 2020), the proposed Large models provided better performances, having faster inference speed. QuickNet-Small showed fast inference speed because QuickNet models were optimized considering the mechanism of LCE (Bannink et al., 2021). However, its performance was only 59.4% Top-1 accuracy on ImageNet-1K.

On the other hand, the experiments with the Samsung Exynos-9820 processor adopted TensorFlow Lite interpreter via Android app. The latencies of the models in Table 3 were measured by averaging the results of 300 runs. In the experiments, all proposed models were faster than the listed FP32 models. Furthermore, the proposed models achieved significant speedups compared to the baseline ReActNetA. Besides, QSB-Net-Large(SE1) provided the highest Top-1 accuracy of 71.1%, having 22 ms inference latency. As shown in Table 3, the main weakness of the proposed models is the increasing storage costs for the final FC layer. The final FC layer consumed about 8 Mbytes storage, which was more than $\frac{2}{3}$ storage costs of the proposed models. As explained in subsection 4.5, the quantization of the FC layer can mitigate the problem, so the values in parentheses in Table 3 prove the reduction of storage costs using the quantization of the FC layer. The effects of quantization on the FC layer in terms of model performance will be discussed in Ablation Studies.

### 5.2 SEMANTIC SEGMENTATION

We evaluated the proposed models on the PASCAL VOC 2012 dataset (Everingham et al., 2010) for semantic segmentation. The detailed information about the dataset is in Appendix A.10. We measured mIoU on 20 object classes and 1 background class, following the training recipe described in DeepLabv3 (Chen et al., 2017). ReActNetA (Liu et al., 2020) had 0.5 mIoU lower result compared with QB-Net-Small. Besides, QSB-Net-Large was the best-performing binarized segmentation model, outperforming FP32 ResNet18 (He et al., 2016) and CBNN (Zhou et al., 2023).

Furthermore, the QSB-Net-Small and Large models significantly enhanced performance compared with LQ-Net (Zhang et al., 2018) using 3-bit quantization.

### 5.3 Ablation Studies

**Deployments of $1 \times 1$ convolutions:** In a modification of QB-Net-Large, when the number of input channels was 16 or 32, binarized pointwise convolutions were adopted after DS convolutions. In this case, the Top-1 accuracy on ImageNet-1K was 68.0%, which was significantly degraded by 1.8%. In another modification of QB-Net-Large, when the number of input channels was 16, 32, or 128, binarized $3 \times 3$ convolutions were adopted after DS convolutions. In this case, its Top-1 accuracy was 70.0%, which was only increased by 0.2%. Based on the above empirical findings, we concluded that when the number of channels in shallow blocks was small, the accuracy can be significantly degraded. Therefore, we determined that instead of binarized pointwise convolutions, binarized $3 \times 3$ convolutions were used when the number of input channels was 16 or 32.

**Effects of $\times 8$ channel expansion in deep blocks:** A modification of QB-Net-Small changed the number of output channels of the last 13-th and 14-th blocks to 4048 ($C = 4048$). The modification produced 69.5% Top-1 accuracy, having 2.6% performance enhancement. This study shows that the increasing complexity of deep blocks can significantly enhance model performance. However, we thought that the storage costs of the FC layer and computational costs of binarized pointwise convolutions were significant, so we did not consider the case with $C = 4048$.

**Learnable bias deployed before DS convolutions:** In a modification of QB-Net-Large, when the learnable bias layer was not deployed just before DS convolutions, Top-1 accuracy was degraded by 2.1% on ImageNet-1K. The above result indicates that the calibration using the learnable bias can be effective in the DS convolutions of the proposed models.

**Quantization of DS convolutions and FC layer:** In a modification of QSB-Net-Large(SE2), we applied 8-bit quantization to the weights and input features for DS convolutions in each block. In the two-stage training, its training result surprisingly achieved up to 71.5% Top-1 accuracy on ImageNet-1K. Then, we applied 8-bit quantization to the weights and input features for the DS convolution and FC layers, where the tuning adopted the learning rate of $10^{-4}$ and 25 epochs on the pretrained model. In the evaluation, Top-1 accuracy was 70.8%, demonstrating that 8-bit quantization was applicable to the proposed models.

**Training from scratch without teacher:** To know the model performance without using the teacher-student training, the proposed models were trained from scratch with the scheduler of cosine annealing with warmup during 600 epochs, which was based on the recipe of QuickNet (Bannink et al., 2021). QB-Net-Large, QSB-Net-Large, and QSB-Net-Large(SE1) had 65.8%, 67.0%, and 67.5% Top-1 accuracies on ImageNet-1K, which indicates that the teacher-student training was critical for enhancing model performance. Whereas the proposed models were slower than Quick-Net (Bannink et al., 2021), Top-1 accuracies of the proposed models without using the teacher-student training significantly outperformed 63.3% Top-1 accuracy of QuickNet in Table 3.

## 6 Conclusion

This paper proposes new BCNNs having low computational costs and high performance using channel quadrupling and smooth downsampling. The proposed structure using the channel quadrupling has low-cost computations in shallow blocks and a wider dynamic range in the frequency domain due to the increased model complexity in deep blocks. The techniques for achieving better models, such as the self-attention (Hu et al., 2018) and the quantization of DS convolutional and FC layers, are applicable to the proposed models. Using multiple binarized convolutions in the FC layer, the proposed models can significantly reduce storage costs without using an 8-bit quantized FC layer. When the self-attention is applied to the DS convolutions during downsampling, its performance on ImageNet-1K reaches up to 71.1% Top-1 accuracy, only requiring $0.63 \times 10^8$ OPs. The small latencies on an RPi 4B and a Samsung Exynos processor prove that the proposed models are suitable for implementing high-speed inference on real hardware. Considering the above performance enhancements and structural benefits, it is concluded that the proposed models are efficient for mobile-based applications. Additional explanations and visualizations of the proposed models are included in Appendixes A.8, A.9, and A.10.

## ETHIC STATEMENT

This paper proposes new binarized convolutional neural network (BCNN) models, introducing novel architectures to enhance model performance and inference speed on real hardware. In this study, we have kept the following ethical principles of ICLR 2025 as:

1. **Contribute to Society and Human Well-being:** The aim of this study is to develop efficient BCNNs by reducing computational costs in shallow blocks and increasing representation capacity in deep blocks. The development of the proposed mobile-friendly models has the potential to benefit a wide range of societal applications, including healthcare, education, and environmental monitoring, particularly in resource-constrained applications.

2. **Uphold High Standards of Scientific Excellence:** We have intensively performed experiments to validate our proposed models. The motivations, ideas, and conclusions are presented to contribute to the scientific community.

3. **Avoid Harm:** The study does not include any human subjects or sensitive personal data. We strongly discourage any misuse of our work that could harm individuals, although there is no explicit information about the misuse in the manuscript.

4. **Be Honest, Trustworthy, and Transparent:** We have honestly reported our research findings, including both strengths and limitations. We sincerely reported the weak points of the proposed models and the method to achieve better models. All data sources, model structures, and experimental environments are fully disclosed to ensure transparency.

5. **Be Fair and Take Action to Avoid Discrimination:** Because we adopt public datasets such as ImageNet-1K and public Pytorch library, experiments can be fair without any discrimination.

6. **Respect the Work Required to Produce New Ideas and Artefacts:** We cite all relevant references in the manuscript to respect existing works. This paper is written considering the previous works and knowledge.

7. **Respect Privacy:** The datasets adopted in our experiments, such as ImageNet-1K and PASCAL VOC, are publicly available and do not contain critical personal information.

8. **Honour Confidentiality:** This paper does not have any confidentiality agreements.

## REPRODUCIBILITY STATEMENT

We adopt conventional ImageNet-1K and PASCAL VOC datasets for easy reproduction. The attached code as supplementary materials can run when Pytorch dataset formats are prepared. The detailed model structures are described in the main body of this paper and Appendix A.1. Detailed explanations of experimental environments and training processes are included in Appendix A.4. Besides, the environments for evaluating inference speed on real hardware are described in Appendix A.7.

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

## A    APPENDIX / SUPPLEMENTAL MATERIAL

### A.1    MODEL STRUCTURES OF QB-NET-LARGE AND QSB-NET-LARGE

**Table 5:** Model structures of QB-Net-Large and QSB-Net-Large for ImageNet-1K.

| Index | QB-Net-Large | | | QSB-Net-Large | | |
|---|---|---|---|---|---|---|
| | Block | Input | Output | Block | Input | Output |
| 1 | Conv | $3\times224\times224$ | $16\times112\times112$ | Conv | $3\times224\times224$ | $16\times112\times112$ |
| 2 | TypeN | $16\times112\times112$ | $16\times112\times112$ | TypeN | $16\times112\times112$ | $16\times112\times112$ |
| 3 | TypeD | $16\times112\times112$ | $32\times56\times56$ | TypeDS | $16\times112\times112$ | $32\times56\times56$ |
| 4 | TypeN | $32\times56\times56$ | $32\times56\times56$ | TypeN | $32\times56\times56$ | $32\times56\times56$ |
| 5 | TypeQ | $32\times56\times56$ | $128\times28\times28$ | TypeQS | $32\times56\times56$ | $128\times28\times28$ |
| 6 | TypeN | $128\times28\times28$ | $128\times28\times28$ | TypeN | $128\times28\times28$ | $128\times28\times28$ |
| 7 | TypeQ | $128\times28\times28$ | $512\times14\times14$ | TypeQS | $128\times28\times28$ | $512\times14\times14$ |
| 8 | TypeN | $512\times14\times14$ | $512\times14\times14$ | TypeN | $512\times14\times14$ | $512\times14\times14$ |
| 9 | TypeQ | $512\times14\times14$ | $2048\times7\times7$ | TypeQS | $512\times14\times14$ | $2048\times7\times7$ |
| 10 | TypeN | $2048\times7\times7$ | $2048\times7\times7$ | TypeN | $2048\times7\times7$ | $2048\times7\times7$ |
| 11 | TypeN | $2048\times7\times7$ | $2048\times7\times7$ | TypeN | $2048\times7\times7$ | $2048\times7\times7$ |
| 12 | TypeN | $2048\times7\times7$ | $2048\times7\times7$ | TypeN | $2048\times7\times7$ | $2048\times7\times7$ |
| 13 | TypeN | $2048\times7\times7$ | $2048\times7\times7$ | TypeN | $2048\times7\times7$ | $2048\times7\times7$ |
| 14 | TypeN | $2048\times7\times7$ | $2048\times7\times7$ | TypeN | $2048\times7\times7$ | $2048\times7\times7$ |
| 15 | AvgPool | $2048\times7\times7$ | $2048\times1\times1$ | AvgPool | $2048\times7\times7$ | $2048\times1\times1$ |
| 16 | FC | $2048\times1\times1$ | $1000$ | FC | $2048\times1\times1$ | $1000$ |

### A.2    BINARIZED CONVOLUTIONS IN FINAL FC LAYER

Let us assume that $x_i$ denotes a feature in input channel $i$. A $n$-bit quantized weight $w_i$ for input channel $i$ is denoted as $w_i = w_i^{n-1}2^{n-1} + w_i^{n-2}2^{n-2} + \cdots + w_i^1 2^1 + w_i^0 2^0$. When $w_i^7, w_i^6, \cdots, w_i^1, w_i^0 \in \{-1, +1\}$, $w_i \in \{-255, -253, \cdots, +253, +255\}$, showing the uniform interval between quantized weights of $w_i$. When we assume $n = 8$ for 8-bit quantized operations, a multiply operation is formulated as $x_i \cdot w_i = w_i^7 x_i 2^7 + w_i^6 x_i 2^6 + \cdots + w_i^1 x_i 2^1 + w_i^0 x_i 2^0$.

When the number of input channels is $C$, an output feature $x_{out}$ of a binarized pointwise convolution can be calculated as:

$$x_{out} = scale \cdot \sum_{i=1}^{C}(w_i^7 x_i 2^7 + w_i^6 x_i 2^6 + \cdots + w_i^1 x_i 2^1 + w_i^0 x_i 2^0), \tag{2}$$

where the index of an output channel is not shown for clarity. Term *scale* is the scaling factor for a binarized convolution. Eq.(2) can be represented as:

$$x_{out} = scale \cdot (2^7 \underbrace{\sum_{i=1}^{C} w_i^7 x_i}_{1\times1\,\text{BCONV}} + 2^6 \underbrace{\sum_{i=1}^{C} w_i^6 x_i}_{1\times1\,\text{BCONV}} + \cdots + 2^1 \underbrace{\sum_{i=1}^{C} w_i^1 x_i}_{1\times1\,\text{BCONV}} + 2^0 \underbrace{\sum_{i=1}^{C} w_i^0 x_i}_{1\times1\,\text{BCONV}}). \tag{3}$$

While the binarized convolution in Rastegari et al. (2016); Bulat et al. (2019) is formulated as $x_{out} = scale \cdot \sum_{i=1}^{C} w_i^0 x_i^0, x_i \in \{-1, +1\}$, 8 binarized convolutions denoted as $1\times1$ BCONV are required in Eq.(3). Considering Eq.(2), the FC layer with 8-bit quantized weights can be decomposed into 8 binarized convolutions. The change of the latency using the decomposed binarized convolutions was negligible on real hardware because the amount of computations in the FC layer was very small. However, the costs for storing the parameters for the FC layer can be significantly reduced. In the proposed QSB-Net, the number of input channels for the FC layer was 2048. When ImageNet-1K was used as the dataset, the FP32 FC layer consumed 8.19 Mbytes in the proposed models. However, when 8 binarized convolutions were deployed in the FC layer, the storage costs were only 2.05 Mbytes, having $\times4$ memory efficiency. When the mixed precision using both FP32 and 8-bit fixed-point operations was not available in the hardware platform for BCNNs, the decomposed binarized convolutions in the FC layer are effective in reducing the total storage costs.

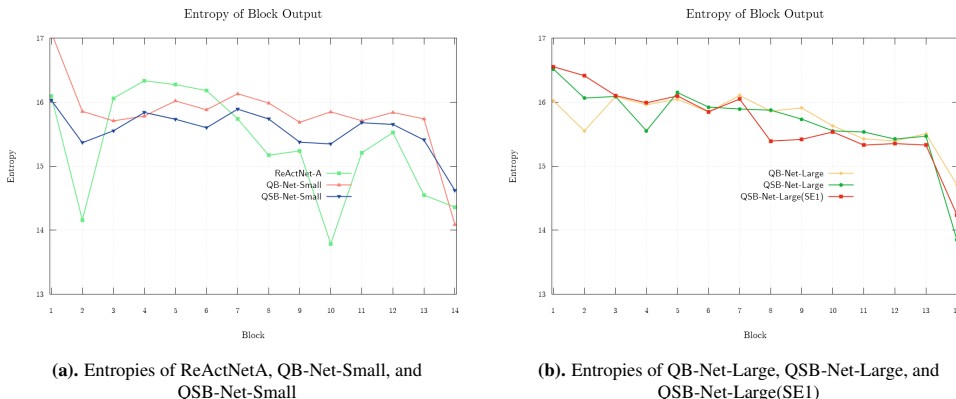

**(a).** Entropies of ReActNetA, QB-Net-Small, and QSB-Net-Small

**(b).** Entropies of QB-Net-Large, QSB-Net-Large, and QSB-Net-Large(SE1)

**Figure 6:** Visualized entropies of block outputs from ReActNetA and proposed models.

### A.3 DETAILED PRESENTATION OF ENTROPY IN PROPOSED MODELS

We think that the downsampling of feature maps and binarization error of BCNNs could affect the information loss. Entropy can be exploited to design the behavior of a specific layer (Wan et al., 2019; Chen et al., 2020b; Zhao & Zhang, 2021). The above existing works motivate us to believe that entropy can be used to explain the information characteristics of output features in BCNNs.

The formula for entropy $H(X)$ is specifically for discrete variable $x_i \in X = \{x_1, x_2, \cdots, x_{n-1}, x_n\}$, which is expressed as follows:

$$H(X) = -\sum_{i=1}^{n} p(x_i) \log_2 p(x_i), \tag{4}$$

where $p(x_i)$ is the probability of each possible value $x_i \in X$. With Eq.(4), the entropy for a block was calculated as: firstly, because the values were continuous, the values of output feature maps in a layer were discretized by binning the values using a histogram. The probability distribution was determined by the relative frequency of each bin from the histogram. Then, the above probability distribution was used to calculate the entropy using the discrete entropy formula in Eq.(4).

We adopted five images from the validation dataset of ImageNet-1K. The images were used as the inputs for the pretrained models. In the inference, the output features were hooked and used to calculate the entropy of each block. Figure 6 (a) illustrates that the entropy of each block significantly fluctuated in baselined ReActNetA, QB-Net-Small, and QSB-Net-Small. On the other hand, Figure 6 (b) shows that the fluctuation of entropy is reduced. We think that the above trends show that feature distributions can change smoothly during feedforwarding. Besides, it is concluded that the increasing representation capacity in deep blocks of QB-Net-Large and QSB-Net-Large can make the change of feature distributions more stable.

### A.4 DETAILED DESCRIPTION OF TRAINING PROCESS IN IMAGE CLASSIFICATION

ImageNet-1K contains 1.3M training and 50K validation images with 1,000 classes. During training on the ImageNet-1K dataset, $224 \times 224$ images based on the augmentations in Liu et al. (2020) were adopted. In inference, $224 \times 224$ center-cropped images from the validation dataset were adopted without any specific image augmentation.

Like other BCNN models, the first convolutional and final FC layers adopted FP32 weights and activations. In order to reduce the storage costs of the final FC layer, 8-bit quantization can be applied. For an apple-to-apple comparison, we adopted ADAM (Kingma & Ba, 2014) optimizer in all cases, having $\beta_1 = 0.9$ and $\beta_2 = 0.999$. When training during $E_{epochs}$ epochs, the initial learning rate $lr_{base} = 0.001$ was assigned. During training, the learning rate $lr$ in the $e_{epochs}$-th epoch decreased based on **poly** policy, which limited the maximum learning rate of the ADAM optimizer (Kingma & Ba, 2014) by $lr_{base} \times (1 - e_{epochs}/E_{epochs})$. In the two-stage training, a teacher-student training method (Hinton et al., 2015) was adopted using the pretrained ResNet34 (He et al., 2016) from Pytorch official site as a teacher.

In order to know the effects of structural benefits, the training recipe of ReActNetA (Liu et al., 2020), which was a well-known BCNN model motivated by MobileNetV1 (Phan et al., 2020a), was adopted. By employing the same training recipe, our analysis demonstrates that the proposed model structures achieve comparable or enhanced performance with reduced computational costs. Based on the above apple-to-apple comparison, we mainly focused on the demonstration of enhancements by adapting model structures. In the first stage with 256 epochs, whereas the input features for BCONVs were binarized, weights were FP32 values. The weight decay in the first stage was set as $10^{-5}$. In the second stage, the pretrained weights from the first stage were used in the initialization. Both input features and weights for BCONVs were binarized during 256 training epochs. The weight decay in the second stage was set as zero. All experiments were conducted on a machine having an AMD Ryzen Threadripper PRO 5955WX 16-Core CPU, 2 NVIDIA RTX 4090 GPUs, and 256 GB RAM. Although the exact training time depended on the status of computing resource usage, the total training times of QSB-Net-Large and QB-Net-Small on our machine were up to about 6 and 7.5 days, respectively. For the approximation of the gradient of sign function

In order to know the effects of the teacher-student training, the proposed models were trained from scratch with the scheduler of cosine annealing with warmup during 600 epochs. In this experiments, we adopted ADAM (Kingma & Ba, 2014) optimizer, having $\beta_1 = 0.9$ and $\beta_2 = 0.999$. The initial learning rate of $lr_{base} = 0.001$ was assigned, having 5 warmup steps with 0.001 maximum learning rate and zero minimum learning rate. After performing warmup steps, the learning rate decreased. The total training times of QB-Net-Large and QSB-Net-Large(SE1) in our machine were up to about 8 and 9 days, respectively.

A.5 DETAIL DISCUSSION FOR ABLATION STUDIES

**Effects of Channel Expansion and Computational Complexity in Shallow and Deep Layers:**
Firstly, by deploying **TypeN** and **TypeD** blocks and considering the channel expansion of Mo-
bileNetV1 and ReActNetA in Table 1, the model performance without channel quadrupling was
investigated. In the above ablation model denoted as **Ablation 1**, the classification on ImageNet
achieved only 64.2% Top-1 accuracy.

In order to know the effects of model complexity in the shallow layer, the following experiments
were performed as: Firstly, in order to know the effects of $3 \times 3$ binarized convolutions in shallow
layers, when the number of output channels 16 and 32, we adopted binarized $1 \times 1$ pointwise con-
volutions in QB-Net-Small instead of $3 \times 3$ binarized convolutions. In the above ablation model
denoted as **Ablation 2**, the classification on ImageNet achieved only 66.7% Top-1 accuracy. Com-
pared with 66.9% Top-1 accuracy of QB-Net-Small in Table 1, there was only 0.2% Top-1 accuracy
drop. Secondly, as shown in Table 6, we evaluated an ablation model by decreasing the number of
channels in the shallow layer, which is denoted as **Ablation 3**. In Ablation 3, the classification on
ImageNet-1K achieved 66.3% Top-1 accuracy. The evaluation in Ablations 2 and 3 indicated that
the model complexity in the shallow layer could not be critical.

In another ablation study, we increased the model complexity in deep layers as: Firstly, instead of
binarized $1 \times 1$ pointwise convolutions, binarized $3 \times 3$ convolutions were used in a modification
of QB-Net-Small when the numbers of output channels were 16, 32, and 128. In the above ablation
model denoted as **Ablation 4**, the classification on ImageNet-1K achieved 67.9% Top-1 accuracy,
which outperformed QB-Net-Small by 1%. On the other hand, when the numbers of output channels
were 16, 32, and 128, a modification of QB-Net-Large adopted binarized $3 \times 3$ convolutions instead
of binarized $1 \times 1$ pointwise convolutions. In the above ablation model denoted as **Ablation 5**, the
classification on ImageNet-1K achieved 70.0% Top-1 accuracy, which outperformed QB-Net-Large
only by 0.2%. Moreover, when a modification of QB-Net-Small denoted as **Ablation 6** had 4096
output channels in the last two depthwise separable convolutions and binarized pointwise convolu-
tions in Table 6, it achieved 69.5% Top-1 accuracy on ImageNet-1K. In the comparison between
Ablations 4, 5, and 6, as the computational complexity in the deeper layers increased, it was con-
cluded that the increasing computational complexity in shallow layers has no significant impact on
the model performance.

**Table 6:** Model structures from ablation studies on ImageNet-1K.

| Index | | Ablation 3 | | | Ablation 6 | |
| --- | --- | --- | --- | --- | --- | --- |
| | Block | Input | Output | Block | Input | Output |
| 1 | Conv | $3 \times 224 \times 224$ | $8 \times 112 \times 112$ | Conv | $3 \times 224 \times 224$ | $16 \times 112 \times 112$ |
| 2 | - | $8 \times 112 \times 112$ | $16 \times 112 \times 112$ | TypeN | $16 \times 112 \times 112$ | $16 \times 112 \times 112$ |
| 3 | TypeD | $16 \times 112 \times 112$ | $32 \times 56 \times 56$ | TypeD | $16 \times 112 \times 112$ | $32 \times 56 \times 56$ |
| 4 | TypeN | $32 \times 56 \times 56$ | $32 \times 56 \times 56$ | TypeN | $32 \times 56 \times 56$ | $32 \times 56 \times 56$ |
| 5 | TypeQ | $32 \times 56 \times 56$ | $128 \times 28 \times 28$ | TypeQ | $32 \times 56 \times 56$ | $128 \times 28 \times 28$ |
| 6 | TypeN | $128 \times 28 \times 28$ | $128 \times 28 \times 28$ | TypeN | $128 \times 28 \times 28$ | $128 \times 28 \times 28$ |
| 7 | TypeQ | $128 \times 28 \times 28$ | $512 \times 14 \times 14$ | TypeQ | $128 \times 28 \times 28$ | $512 \times 14 \times 14$ |
| 8 | TypeN | $512 \times 14 \times 14$ | $512 \times 14 \times 14$ | TypeN | $512 \times 14 \times 14$ | $512 \times 14 \times 14$ |
| 9 | TypeN | $512 \times 14 \times 14$ | $512 \times 14 \times 14$ | TypeN | $512 \times 14 \times 14$ | $512 \times 14 \times 14$ |
| 10 | TypeN | $512 \times 14 \times 14$ | $512 \times 14 \times 14$ | TypeN | $512 \times 14 \times 14$ | $512 \times 14 \times 14$ |
| 11 | TypeN | $512 \times 14 \times 14$ | $512 \times 14 \times 14$ | TypeN | $512 \times 14 \times 14$ | $512 \times 14 \times 14$ |
| 12 | TypeN | $512 \times 14 \times 14$ | $512 \times 14 \times 14$ | TypeN | $512 \times 14 \times 14$ | $512 \times 14 \times 14$ |
| 13 | TypeQ | $512 \times 14 \times 14$ | $2048 \times 7 \times 7$ | - | $512 \times 14 \times 14$ | $4096 \times 7 \times 7$ |
| 14 | TypeN | $2048 \times 7 \times 7$ | $2048 \times 7 \times 7$ | TypeN | $4096 \times 7 \times 7$ | $4096 \times 7 \times 7$ |
| 15 | AvgPool | $2048 \times 7 \times 7$ | $2048 \times 1 \times 1$ | AvgPool | $4096 \times 7 \times 7$ | $4096 \times 1 \times 1$ |
| 16 | FC | $2048 \times 1 \times 1$ | 1000 | FC | $4096 \times 1 \times 1$ | 1000 |

**Effects of Smooth Downsampling:** As shown in Figure 3, the smooth downsampling was only
performed in **TypeDS** and **TypeQS**, so that the effects of smooth downsampling can be discussed
with the performance gap between QB-Net and QSB-Net. In a modification of Ablation 1 having
the channel expansion of MobileNetV1 and ReActNetA, the TypeD block was applied. In the above
modification denoted as **Ablation 7**, Top-1 accuracy was enhanced by 70.5%, which showed that

the smooth downsampling is effective in both the existing baseline and proposed models. However, the computational costs were greater than those of ReActNetA due to the additional computation from the smooth downsampling. In the comparison between QB-Net and QSB-Net models, QSB-Net-Small and QSB-Net-Large enhanced Top-1 accuracy by 1.9% and 0.8% over QB-Net-Small and QB-Net-Large, respectively. The above experimental results indicated that as the computational complexity in deep layers increased, the effects of the smooth downsampling were limited. The below Table 7 summarizes the performance of the ablation models in the image classifications on ImageNet-1K.

**Table 7:** Summary of ablation studies with additional experiments on ImageNet-1K.

| Index | Modification | Top-1(%) | Notice |
|---|---|---|---|
| 1 | Ablation 1 | 64.2 | Channel expansion from MobileNetV1 and ReActNet without smooth downsampling |
| 2 | Ablation 2 | 66.7 | Binarized $1 \times 1$ convolutions for # channels with 16 and 32(QB-Net-Small) |
| 3 | Ablation 3 | 66.3 | 8 output channels for the first convolution(QB-Net-Small) |
| 4 | Ablation 4 | 67.9 | Binarized $3 \times 3$ convolutions when # output channels are 16, 32, and 128 (QB-Net-Small) |
| 5 | Ablation 5 | 70.0 | Binarized $3 \times 3$ convolutions when # output channels are 16, 32, and 128 (QB-Net-Large) |
| 6 | Ablation 6 | 69.5 | 4096 output channels in the last two DS layers(QB-Net-Small) |
| 7 | Ablation 7 | 70.5 | Modification of Ablation 1 with smooth downsampling |
| 8 | QB-Net-Small | 66.9 | Small computational complexity in deep layers without smooth downsampling |
| 9 | QSB-Net-Small | 68.8 | Small computational complexity in deep layers with smooth downsampling |
| 10 | QB-Net-Large | 69.8 | Large computational complexity in deep layers without smooth downsampling |
| 11 | QSB-Net-Large | 70.6 | Large computational complexity in deep layers with smooth downsampling |

## A.6 Experimental Results on COCO-Stuff

To demonstrate the generalization ability of the proposed models, we conducted semantic segmentation experiments on the large-scale dataset COCO-Stuff Caesar et al. (2018). In Table 8, QSB-Net-Large and QB-Net-Large achieved higher performance than the FP32 precision models with a similar number of parameters while using 1-bit weights and features.

**Table 8:** Comparison of semantic segmentation on COCO-Stuff dataset.

| Model | Params | W/F | mIoU |
|---|---|---|---|
| BiSeNetv2-L (Yu et al., 2021) | - | 32/32 | 28.7 |
| DDRNet23 (Hong et al., 2021) | 20.1M | 32/32 | 32.1 |
| PSPNet50 (Zhao et al., 2017) | - | 32/32 | 32.6 |
| RTFormer-B (Wang et al., 2022) | 16.8M | 32/32 | 35.3 |
| **QB-Net-Large** | 12.0M | 1/1 | 36.4 |
| **QSB-Net-Large** | 12.3M | 1/1 | 37.5 |

Notably, QSB-Net-Large outperformed the Transformer-based RTFormer-B (Wang et al. (2022)) by 2.2 mIoU. It can be seen that the proposed model has good generalization performance even on large datasets and has a high potential for practical applications.

Besides, object detection was performed on the COCO-Stuff dataset. Table 9 lists the comparison of object detection, where QSB-Net-Large has achieved higher performance than the FP32 precision lightweight models, using 1-bit weights and features. The results show that for vision tasks such as classification, semantic segmentation, and object detection, the proposed model outperformed the accuracy of FP32 precision models.

**Table 9:** Comparison of objection detection on COCO-Stuff dataset.

| Model | W/F | mAP |
|---|---|---|
| Fast-RCNN (Girshick, 2015) | 32/32 | 19.7 |
| LeYOLO-Nano (Hollard et al., 2024) | 32/32 | 25.2 |
| MnasFPN (MobileNetv3) (Chen et al., 2020a) | 32/32 | 25.5 |
| YOLOX-Nano (GeZ et al., 2021) | 32/32 | 25.8 |
| ESPNetv2 (Mehta et al., 2019) | 32/32 | 26.0 |
| **QSB-Net-Large** | 1/1 | 26.4 |

Considering the above semantic segmentation and object detection on the CoCo-Stuff dataset, it is concluded that the proposed models can show an important achievement that overcame the limitations of BCNNs and enhanced its applicability in real applications.

## A.7 EXPERIMENTAL ENVIRONMENTS ON REAL HARDWARE: LARQ

Target models were prepared using TensorFlow Keras framework. XNOR-Net (Rastegari et al., 2016), Real-to-Bin (Martinez et al., 2019), Bi-RealNet (Liu et al., 2018), and QuickNet (Bannink et al., 2021) Keras models were from Larq Zoo (Bannink et al., 2021). The proposed models and other counterparts were coded based on the original Keras layers. The models can be converted into TFLite (TensorFlow Lite) filebuffer files. When checking the inference speed of a model, we adopted Larq Compute Engine (LCE) (Bannink et al., 2021), which provided a benchmark evaluation program based on TensorFlow Lite (ten, 2023) and customized binarized convolutional layers. The benchmark evaluation program ran on Manjaro 64-bit GNOME Desktop for an RPi 4B and Android app for a Samsung Exynos-9820 processor. It was known that LCE provided a collection of hand-optimized TFLite custom operators. Along with the full support of existing TFLite operators, each binarized convolutional layer can be performed using its custom binarized convolution. In our evaluations, the program showed the averaged latencies of 50 runs on the RPi 4B and 300 runs on the Samsung Exynos-9820 processor with randomly generated inputs.

## A.8 Visualization of Feature Maps

Figures 7 and 8 visualize the first five feature maps from the downsampling blocks. The pretrained model of the baseline ReActNetA (Liu et al., 2020) was downloaded from its official GitHub. Compared with ReActNetA, the visualization shows a significant difference in the first and last downsampling blocks. In QSB-Net models, the output feature maps after performing heightwise downsampling are illustrated. The feature maps are denoted as the output of $stride = (2, 1)$. Compared with the feature maps after widthwise downsampling, the output feature maps after heightwise downsampling also show the diversity of features after heightwise downsampling, which could indicate that smooth downsampling in QSB-Net can enhance the representation capacity. In general, the visualizations of the proposed models in Figures 7 and 8 show the diversity of features and increasing representation capacity of deep blocks during downsampling. Also, the visualization in Figures 7 and 8 proves the wide dynamic range in the frequency domain of deep blocks illustrated in Figure 5.

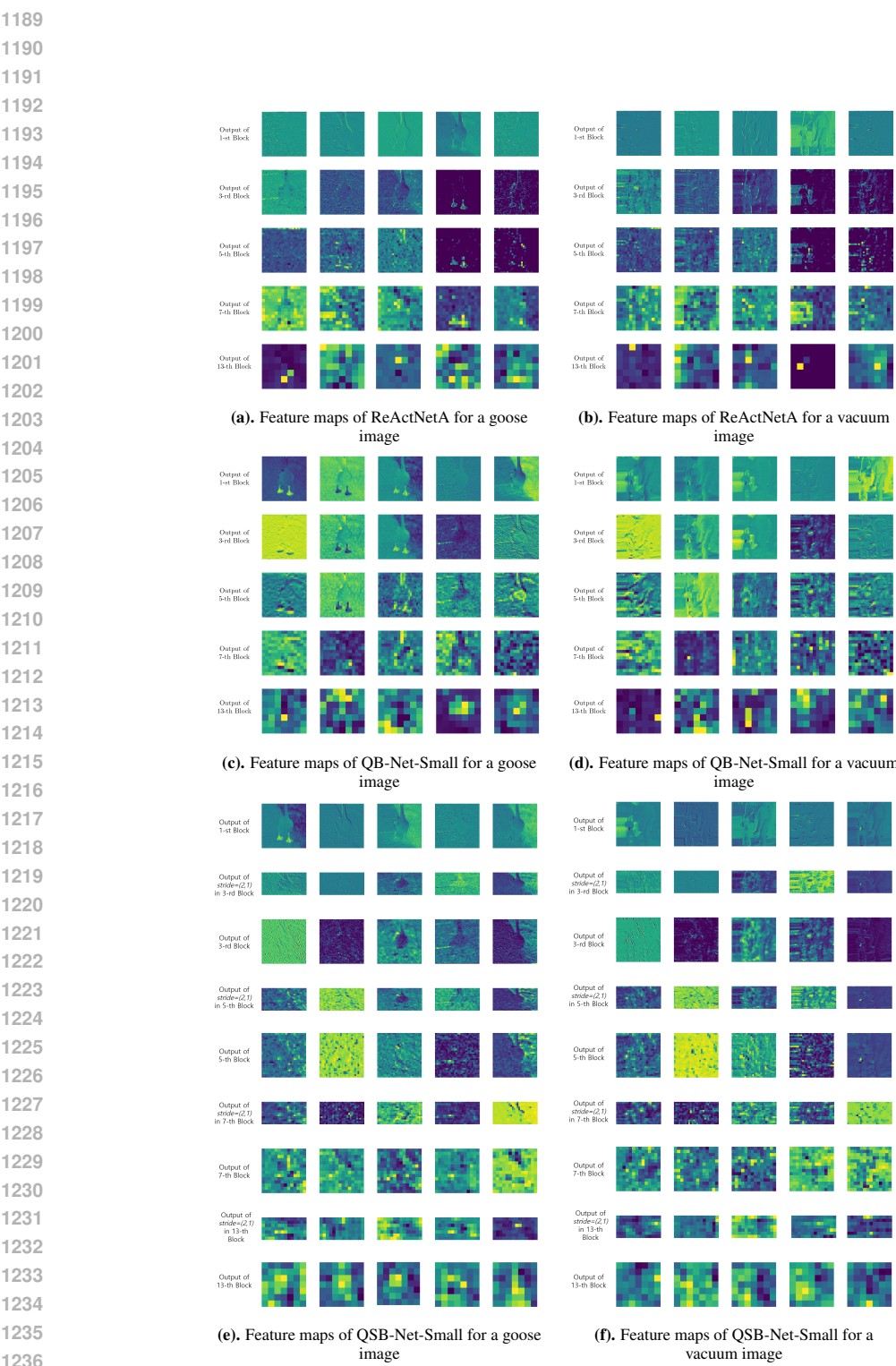

**Figure 7:** Visualizations of feature maps for ReActNetA and proposed Small models.

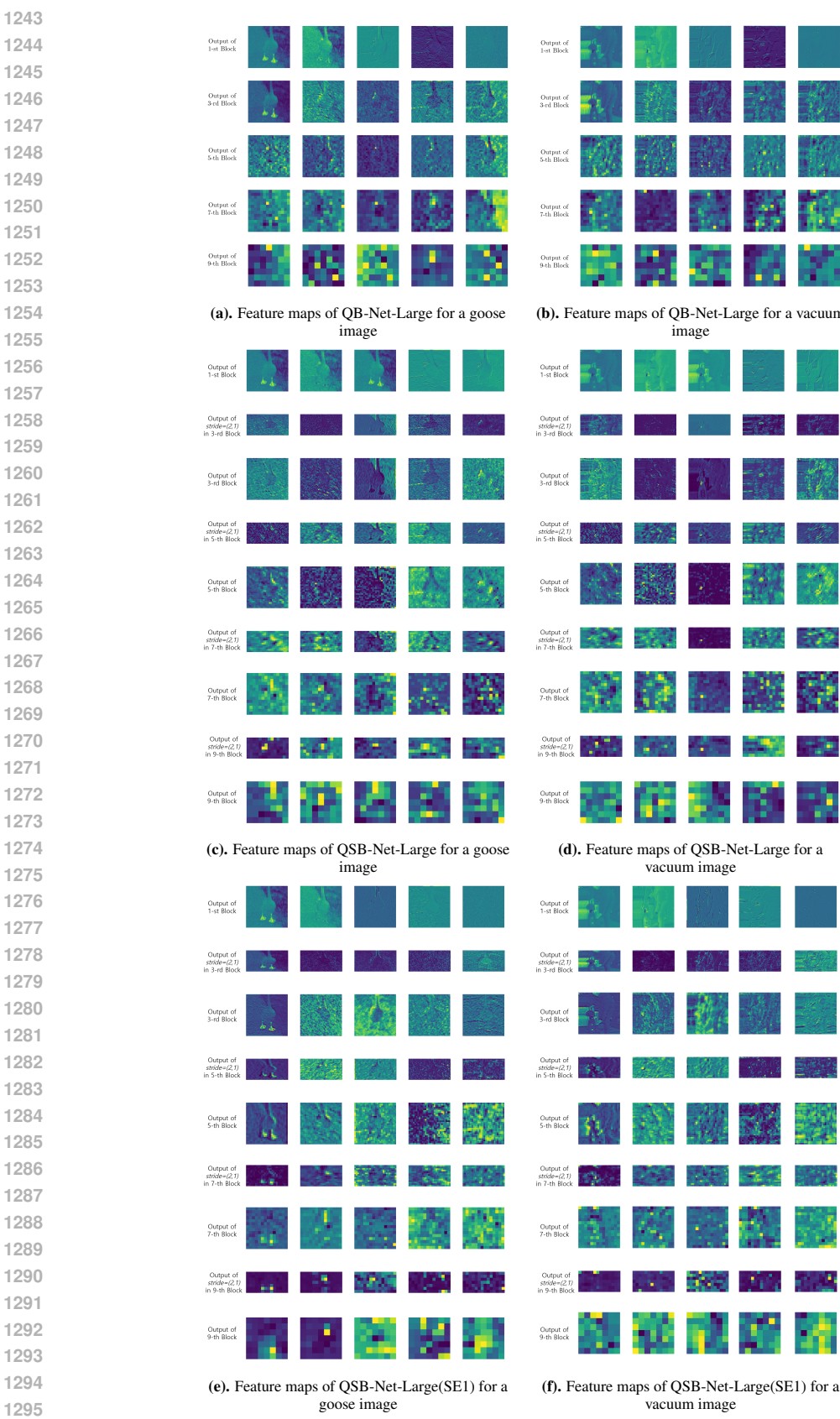

(a). Feature maps of QB-Net-Large for a goose image

(b). Feature maps of QB-Net-Large for a vacuum image

(c). Feature maps of QSB-Net-Large for a goose image

(d). Feature maps of QSB-Net-Large for a vacuum image

(e). Feature maps of QSB-Net-Large(SE1) for a goose image

(f). Feature maps of QSB-Net-Large(SE1) for a vacuum image

**Figure 8:** Visualizations of feature maps for proposed Large models.

## A.9 T-SNE AS A TOOL FOR VISUALIZING AND INTERPRETING IMAGE CLASSIFICATION RESULTS

The evaluations of models using t-distributed stochastic neighbor embedding (t-SNE) are illustrated in Figure 9. In FP32 ResNet18 and MobileNetV2, the visual patterns show that the clustered features are well separated. The visual patterns of ReActNetA (Liu et al., 2020) and proposed models indicate that BCNNs suffer from difficulties in distinguishing between several classes. Compared with the Small models, the Large models slightly reduced the intersection of clustered features. Therefore, we conclude that the visualized patterns indirectly prove the performance enhancements in the Large models.

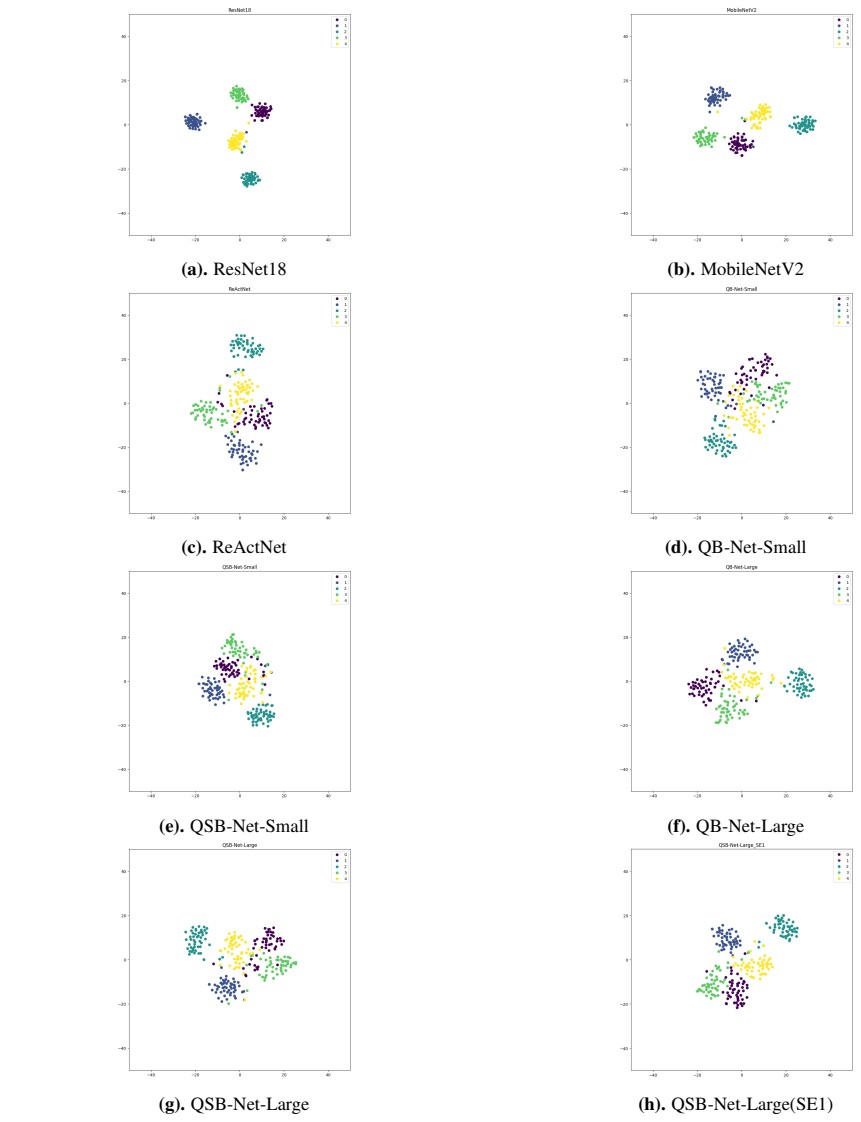

**(a).** ResNet18                    **(b).** MobileNetV2

**(c).** ReActNet                    **(d).** QB-Net-Small

**(e).** QSB-Net-Small                    **(f).** QB-Net-Large

**(g).** QSB-Net-Large                    **(h).** QSB-Net-Large(SE1)

**Figure 9:** Visualizations of t-SNE.

## A.10 VISUALIZED RESULTS OF SEMANTIC SEGMENTATION

We demonstrated that QSB-Net-Large could learn a good representation of objects in semantic segmentation, where QSB-Net-Large combined with DeepLabv3+ was trained. The PASCAL VOC 2012 (Everingham et al., 2010) segmentation dataset contains 1,465 training, 1,449 validation, and 1,456 test images having pixel-level annotations. The dataset was augmented by the extra annotations from the PASCAL VOC 2011, resulting in 10,582 augmented training images. Figures 10 and 12 show the visualized results of semantic segmentation using QSB-Net-Large on the VOC PASCAL 2012 dataset (Everingham et al., 2010).

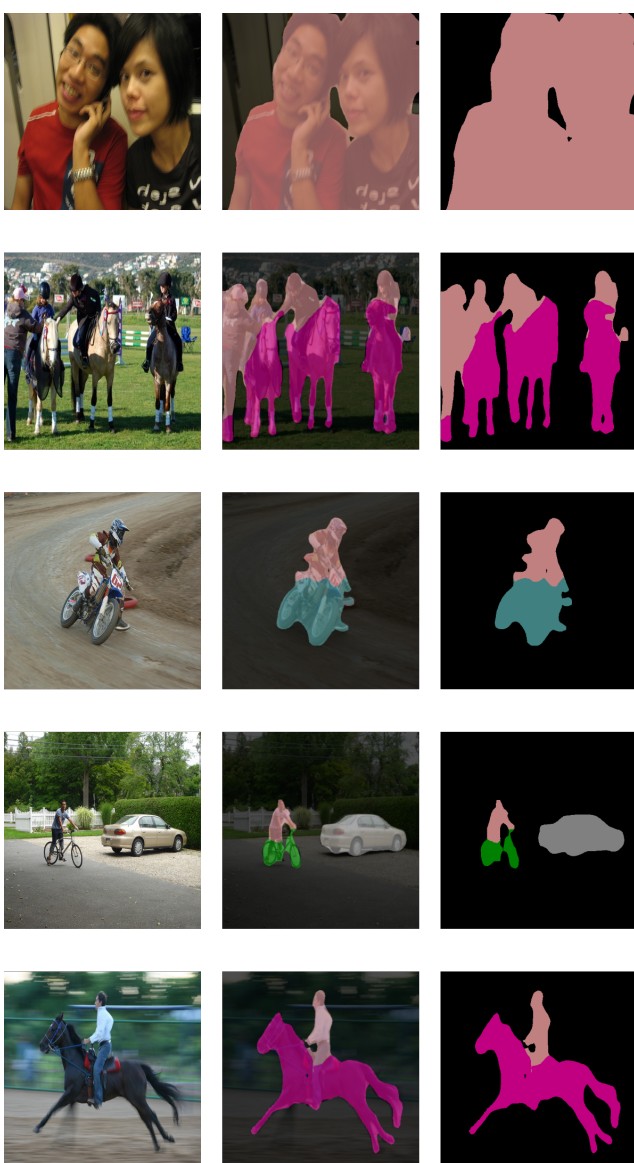

**Figure 10:** Semantic segmentation results of QSB-Net-Large. On the left, input images are illustrated. In the middle, the segmentation masks overlayed on its input image are shown. On the right, the predicted segmentation masks are given.

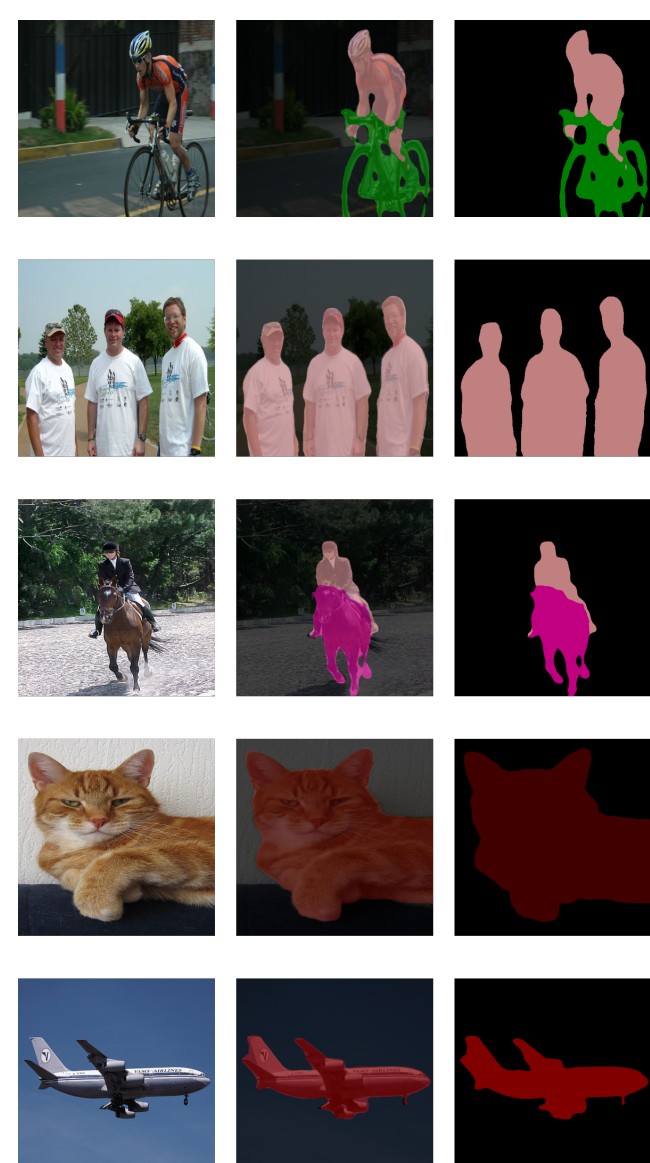

**Figure 11:** Semantic segmentation results of QSB-Net-Large. On the left, input images are illustrated. In the middle, the segmentation masks overlayed on its input image are shown. On the right, the predicted segmentation masks are given.

### A.11   ESTIMATION OF ENERGY CONSUMPTION ON REAL HARDWARE

We performed an energy consumption analysis on real hardware using an RPi 4B on Manjaro OS. Due to the limitations of using the power management ICs (PMIC), we cannot use power profiling tools. Therefore, we employed an alternative method, where we measured the current drawn from the USB power line using a current probe over 500 test runs. To assess the impact of running a benchmark evaluation program in Appendix A.7, we compared the current consumption under two conditions: with and without executing the program on the RPi 4B. When the models were not being evaluated, the current consumption was approximately 713 mA @ 5V. In contrast, during the execution of the models listed in Table 3, the current consumption ranged from 912 mA to 936 mA. Considering the resolution of the current probe, we thought that these variations were not significant.

While other system activities such as parameter uploading and log dumping were also performed during the measurements, their impact on the comparison was not great, considering the negligible

relative difference. Based on these observations, we conclude that the systematic energy consumption is largely proportional to inference latency because no substantial differences in system power consumption were detected during the evaluations.

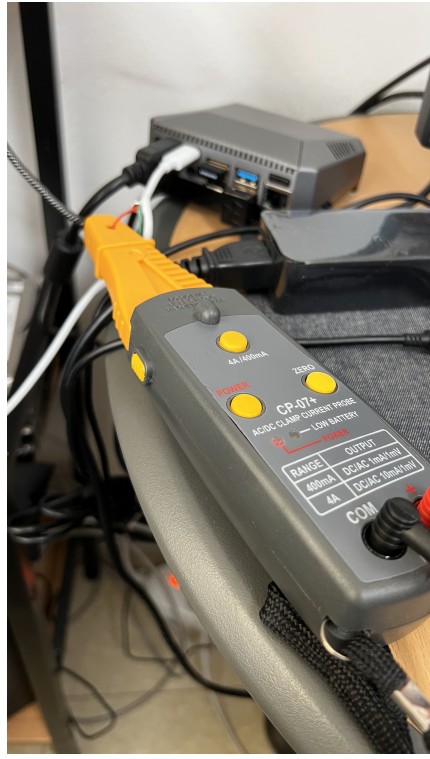

**Figure 12:** A realistic environment for estimating energy consumption. Using a current probe with $\pm$ 1.5% accuracy resolution, the amount of current in a USB power cable was measured.

