# OpenReview forum: "Binarized Convolutional Neural Networks with Channel Quadrupling and Smooth Downsampling"
_ICLR.cc/2025/Conference — Submitted to ICLR 2025_

### Official Review · Reviewer_d56q · 2024-10-27

**Soundness:** 3
**Presentation:** 3
**Contribution:** 2
**Rating:** 5
**Confidence:** 2

**Summary:**

This paper introduces two novel binarized convolutional neural network (BCNN) architectures, QB-Net and QSB-Net, which leverage CHANNEL QUADRUPLING and SMOOTH DOWNSAMPLING to optimize performance in resource-constrained settings. The proposed models integrate FP32 depthwise separable (DS) convolutions with binarized 1x1 pointwise convolutions. To manage model complexity effectively, both models use narrow channels in initial layers and expand channels by a factor of four during downsampling. Additionally, they introduce a smooth downsampling technique that progressively doubles the channel count at each stage. Further, the design incorporates channelwise self-attention (SE) with minimal computational overhead. Experimental results on ImageNet-1K and PASCAL VOC show that QB-Net and QSB-Net achieve superior performance among binarized networks.

**Strengths:**

- The integration of channel quadrupling and smooth down-sampling within BCNNs represents a new approach that effectively balances computational efficiency with model performance.
- The reported accuracy on ImageNet-1K and PASCAL VOC datasets substantiates the efficacy of the proposed models.
- The latency measurements conducted on both the Raspberry Pi 4B and Samsung Exynos-9820 processor demonstrate the practical feasibility of the models in real-world applications.
- The paper offers comprehensive descriptions of the model architectures, experimental setups, and training protocols, enhancing transparency and supporting reproducibility.

**Weaknesses:**

# Major Concern:
While the results presented are promising, the novelty of this paper appears somewhat limited. The model design heavily relies on a handcrafted architecture coupled with a brute-force search for optimizing the binary (or, more accurately, mixed-precision) network architecture. Given the current pace of advancements in the field, this approach may offer limited benefit to the broader community and does not fully align with the novelty standard typically expected at a top-tier machine learning conference.

# Minor Comments:

- The proposed models require over 500 epochs to converge, which is considerably longer than the conventional training time on the ImageNet dataset. To provide a more complete comparison with other binary and full-precision models, it would be beneficial to analyze the impact of these extended training requirements and the use of a teacher model on final model performance.
- Although the proposed models incorporate a range of established techniques through channel quadrupling and smooth down-sampling, the paper would benefit from additional ablation studies to clarify the individual contributions of these primary techniques to model performance.
- Additional visualizations of feature maps could be helpful in illustrating the effects of the proposed smooth down-sampling technique. This would also help clarify whether the observed performance improvements stem from the design itself or from added computational complexity.
- The paper’s primary focus is on image classification tasks, with limited experimental results on the VOC dataset for semantic segmentation. A broader range of experiments, particularly on larger and more diverse datasets (e.g., COCO detection or segmentation tasks), would help to more convincingly demonstrate the generalizability of the proposed model.
- It appears that more recent works on lightweight network design are missing from the comparisons in Table 3. Including these works would provide a more comprehensive context for evaluating the proposed model’s contributions.

**Questions:**

- Could the paper provide a more extensive set of comparative experiments across a broader range of visual tasks and larger-scale datasets (e.g., COCO detection or segmentation tasks) to further validate the generalizability of the proposed model?
- Would it be possible to analyze and compare the impact of training costs on the final performance between the proposed models and existing binary or full-precision models? This would add valuable insight into the efficiency of the proposed approach.
- Could the authors include ablation studies to illustrate the individual contributions of channel quadrupling and smooth down-sampling to the overall model performance? This would clarify the significance of these techniques within the model design.

---

> ### Author Response · Authors · 2024-11-20
> **1st Rebuttal for Reviewer d56q (1)**
>
> Dear Reviewer d56q,
> Thanks for your valuable comments and idea. Considering your comments and ideas, we have prepared rebuttals, which address weak points and answer your question as follows:
>
> ---
>
>  * **Q1 (Major Concern):** Concerns about the handcrafted architecture of the model.
>  * **A1 (Major Concern):** Thank you for your thoughtful feedback. While some design decisions were guided by empirical observations, our work is rooted in a clear and theoretically motivated hypothesis: that information loss during downsampling is a fundamental bottleneck in BCNNs. **To address this, we introduced channel quadrupling and advanced downsampling techniques, which are not merely handcrafted heuristics but are supported by the sophisticate premise that increasing the representation capacity in deeper layers preserves richer representation capacity.** This approach enhances the model's ability to achieve smoother entropy and a broader dynamic range in frequency—a critical advancement we have substantiated through logical analysis.
> &nbsp;&nbsp; &nbsp;&nbsp; Regarding design decisions, we intentionally aimed to reduce the complexity of shallow layers for a lightweight architecture. However, determining how much to reduce this complexity is inherently dataset-dependent, which needs an empirical approach. This is not unique to our work; many existing models could rely on experimental fine-tuning to determine aspects like the number of channels. (One or two channels cannot be used in well-known vision dataset!!)
> &nbsp;&nbsp; &nbsp;&nbsp; Additionally, our design choice aimed at the balance between performance and efficiency by employing FP32 in depthwise separable convolutions and binarized operations in pointwise convolutions for reducing computational complexity, **We have provided a structure optimized for lightweight deployment rather than relying on purely handcrafted heuristics.**
> **This work not only demonstrates practical improvements but also provides new insights into how structural adaptations in BCNNs can enhance model performance, showing the way for more lightweight and generalizable designs.**
> &nbsp;&nbsp; &nbsp;&nbsp; We believe these contributions reflect a carefully considered hypothesis regarding the relationship between information loss and architecture design. Thanks for your concerns.
> ---
>
>  * **Q2 (Minor Comments-1, Question-2):**  Discussion of the impact of training requirements and the use of a teacher model.
>  *   **A2 (Minor Comments-1, , Question-2):**  Thank you for your insightful comments. In the evaluation process, we adopted the training recipe of ReActNetA, a well-known BCNN derived from MobileNetV1, as detailed in Section 5.1. By employing the same training recipe, our analysis demonstrates that the proposed model structures achieve comparable or enhanced performance with reduced computational costs. Based on the above apple-to-apple comparison, we mainly focused on the demonstration of enhancements by adapting model structures.  **In the revised version, the above explanation has been added in A.4. Detailed Description of Training Process in Image Classification.**
> &nbsp;&nbsp; &nbsp;&nbsp;The training recipe can be categorized into quantization-ware training (QTA), which was known that it took longer training time. Besides, a straight-through estimator (STE) was adopted for the derivatives of sign function in our works, which needed slow convergence due to less accurate gradient information. Unlike the STEs, the complex derivatives was adopted in ReActNet. However, the complex derivatives significantly increased training time in our evaluation, which as written in line 157-162.
> &nbsp;&nbsp; &nbsp;&nbsp;Although the same recipe with ReActNetA was adopted without using the customized sign function, the proposed models showed good performances with reduced computational costs. **Considering the reduced training overhead and performance efficiency, we assure that the proposed models are beneficial.**
> &nbsp;&nbsp; &nbsp;&nbsp;Additionally, in the training-from-scratch experiments presented in Section 5.3 (Ablation Studies), we followed the training recipe of QuickNet. Many existing BCNN models have their unique training recipes, and we adopted these two established training recipes to evaluate the structural advantages and the impact of teacher-student training.
> &nbsp;&nbsp; &nbsp;&nbsp;**The above results clearly indicated that teacher-student training played a critical role in the performance of our proposed models.** For instance, QB-Net-Large achieved 69.8% Top-1 accuracy on ImageNet-1K with teacher-student training, whereas training from scratch yielded 65.8% Top-1 accuracy—a substantial gap. This finding was emphasized in the fourth section of Section 5.3, where we highlighted that teacher-student training was a critical method in the performance improvement of the proposed models.

---

> ### Author Response · Authors · 2024-11-20
> **1st Rebuttal for Reviewer d56q (2)**
>
> * **Q3 (Minor Comments-2, Question3):**  Ablation studies of channel quadrupling and smooth downsampling.
>  *   **A3 (Minor Comments-2, Question3):** Thanks for your sharp points. We are sorry that you feel the model architecture optimization hasn't been sufficiently analyzed. However, we think that our structural analysis were considered in the manuscript as:
> &nbsp;&nbsp; &nbsp;&nbsp; **Firstly,**  based on the hypothesis that the high representation capacity during downsampling in deep layer can be more beneficial, different types of models (Large and Small models of QB-Net and QSB-Net) were analyzed, where the block structures were given to know the detail model architectures and applicability of self-attention in BCNNs. As shown in Table 1 and Table 5 of Appendix, the different model structures based on our hypothesis were given. We think that **the above analysis can show the relationship between the proposed model architectures and performance.**
> &nbsp;&nbsp; &nbsp;&nbsp;**Secondly, for the detail discussion for effects of channel expansion and computational complexity in shallow and deep layers and smooth downsampling, Appendix A.5 Detail Discussion for Ablation Studies have been added in the revised draft.**  Channel quadrupling was difficult to analyze using simple ablation studies, such as removing individual layers.
> &nbsp;&nbsp; &nbsp;&nbsp;We hope that the reviewers recognize that various ablation studies have been conducted on channel quadrupling and smooth downsampling to validate their contributions.
>
> ---
>  * **Q4 (Minor Comments-3):**  Additional visualizations of feature maps for the smooth down-sampling technique.
>  *   **A4 (Minor Comments-3):**  According to your opinion, **additional visualizations of feature maps with smooth downsampling have been added in Figures 7 and 8 of Appendix A.8** in the revised draft, including brief explanation as:
> &nbsp;&nbsp; &nbsp;&nbsp;**"*In QSB-Net models, the output feature maps after performing heightwise downsampling are illustrated. The feature maps are denoted as the output of $stride=(2,1)$. Compared with the feature maps after widthwise downsampling, the output feature maps after heightwise downsampling also show the diversity of features after heightwise downsampling, which could indicate that smooth downsampling in QSB-Net can enhance the representation capacity.*"**
>
> ---
>  * **Q5 (Minor Comments-4, Question-1):**  Additional experiments on larger-scale datasets such as COCO detection dataset to further validate the generalizability of the proposed model.
>  *   **A5 (Minor Comments-4, Question-1):**   According to your recommendation, the training on the COCO dataset has been performed, where **the training results of QSB-Net-Large and QB-Net-Large have outperformed the recent FP32 models such as RTFormer-B[1], as shown in newly added Appendix A.6 Experimental Result of CoCo Stuff.**
> &nbsp;&nbsp; &nbsp;&nbsp;[1] Jian Wang, Chenhui Gou, Qiman Wu, Haocheng Feng, Junyu Han, Errui Ding, and Jingdong Wang.Rtformer: Efficient design for real-time semantic segmentation with transformer. Advances in Neural Information Processing Systems, 35:7423?7436, 2022.
>
> ---
>  * **Q6 (Minor Comments-5, Question-2):**  More recent works of the comparisons in Table 3.
>  *   **A6 (Minor Comments-5, Question-2):**  Thanks for your good points. We have added the evaluation results of Real-to-Bin(ResNet18) on RPi 4B in Table 3 in the newly modified version. However, the supported layers on real hardware were limited in several models such as RB-Net (reshaped pointwise convolution), AdaBin(Maxout), PokeBNN (8-bit and 4-bit quantization), etc.
> &nbsp;&nbsp; &nbsp;&nbsp; Besides, several model architecture had the same because their contributions were only in the training methods such as SA-BNN and ReCU.  Therefore, as shown in the explanation in the previous draft, the comparison in Table 3 was limited, which was explained in lines 457-458.
> &nbsp;&nbsp; &nbsp;&nbsp; ***"We note that the supported layers in LCE were limited, so only several mobile-friendly CNNs and BCNNs based on ResNet18 and MobileNetV1 were compared in Table 3."***
> &nbsp;&nbsp; &nbsp;&nbsp; Although all models in Table 2 cannot be listed in Table 3, the results of Table 3 show the effectiveness of the proposed models on real hardware.
>
> ---
> Thanks all. Your valuable comments are helpful for enhancing the quality of this paper.

---

> ### Author Response · Authors · 2024-11-24
> **Reminder from Authors regarding the Discussion Period**
>
> Dear Reviewer,
>
> We would like to kindly remind you that we submitted our responses to your comments last Wednesday. As the discussion period is approaching its end, we sincerely ask for your attention to review our response. If you find that our answers sufficiently address your concerns and questions, we would be grateful if you could consider revising your rating.
>
> Thank you very much for your time and effort, and we truly appreciate your valuable feedback.

---

> ### Author Response · Authors · 2024-11-27
> **Additional Information about Q4: Object detection on COCO dataset**
>
> Dear Reviewer d56q,
>
> To address the concerns raised, additional object detection has been performed on the COCO dataset. We apologize for the delay in providing updates due to the extended time required for model training. Notably, when carefully selected counterparts have been evaluated, **the proposed QSB-Net-Large has demonstrated performance surpassing that of many existing FP32 models, despite being a binarized model,** as shown in the table below.
>
> Table. Comparison of object detection on COCO-Stuff dataset
> | Model                 |  W/F  |  mAP |
> |-----------------------|:-----:|:----:|
> | Fast-RCNN             | 32/32 | 19.7 |
> | LeYOLO-Nano           | 32/32 | 25.2 |
> | MnasFPN (MobileNetv3) | 32/32 | 25.5 |
> | YOLOX-Nano            | 32/32 | 25.8 |
> | ESPNetv2              | 32/32 | 26.0 |
> | QSB-Net-Large         |  1/1  | 26.4 |
>
>
> In this Table, QSB-Net-Large has achieved higher performance than the FP32 precision lightweight models, using 1-bit weights and features. **The results show that for vision tasks such as classification, semantic segmentation, and object detection, the proposed model have outperformed the accuracy of FP32 precision models.**
>
> We hope that the additional experiments contribute to further validating the generalizability of the proposed model. If our additional experiments have adequately addressed the reviewers' concerns, we kindly ask you to consider this when evaluating the rating.
> We would like to once again thank the reviewers for their feedback. Since the discussion period is still ongoing, we hope to continue the discussion and address any remaining concerns.

---

> > ### Comment · Reviewer_d56q · 2024-11-28
> > **RE: Additional Information about Q4: Object detection on COCO dataset**
> >
> > I would like to express my sincere gratitude for conducting additional experiments on the COCO dataset. Your efforts are greatly appreciated. However, I have a concern regarding the setting of QSB-Net-Large as 1/1 in terms of W/F. Specifically, I am uncertain whether employing FP32 in depthwise separable convolutions is a suitable representation for this setting. Thank you once again for your detailed elaboration.

---

> ### Author Response · Authors · 2024-11-27
>
> Dear Reviewer d56q and other Reviewers,
>
> As the deadline for finalizing the revised version of the paper's PDF approaches, we have updated the PDF by adding a table presenting the **object detection results on the CoCo-Stuff dataset to Appendix A.6**, as previously mentioned. Since further modifications to the PDF are not possible, if reviewers raise additional discussions or concerns, we will do our best to explain them clearly through the response letter. To address the concerns raised by the five reviewers, several appendices have been added, and revisions and additions have been made to the figures and tables. Please review these sections, and **if you find that the concerns you raised have been adequately addressed, we kindly ask you to reconsider your rating.**
>
> Once again, we sincerely thank the reviewers for their valuable feedback.

---

> ### Author Response · Authors · 2024-11-28
> **Answer for Reviewer d56q's Comments about Additional Information about Q4: Object detection on COCO dataset**
>
> Dear Reviewer d56q,
>
> Thank you so much for your positive feedback regarding the additional experiments conducted on the COCO dataset. In the table for QSB-Net-Large, *W/F* indicates that we mainly binarized the 1x1 pointwise convolution. The other counterpart, where *W/F=32/32*, represents the FP32 model.
> As mentioned in the main text, in a modified version of the well-known MobileNetV1 such as ReActNetA, it replaced the FP32 depthwise separable convolution with a binarized 3x3 convolution. **Considering that depthwise separable convolution has inherently low computational costs, we explained that this part could be implemented as FP32 or, for further lightweight design, as 8-bit integer operations without performance degradation.**
>
> Although existing studies like MoBiNet[1] explored binarizing depthwise separable convolution, **they experienced a significant performance drop exceeding 10% compared to FP32 models.** The goal of our model is to reduce the amount of operations while maintaining high performance, making depthwise separable convolution an better choice.
>
> Once again, thank you for your positive evaluation of our additional experiments. **If this explanation addresses your concerns, we would greatly appreciate it if you could reconsider your rating.** Thank you.
>
> [1] Hai Phan, Yihui He, Marios Savvides, Zhiqiang Shen, et al. Mobinet: A mobile binary network for image classification. In The IEEE Winter Conference on Applications of Computer Vision, pp.
> 3453–3462, 2020a.

---

### Official Review · Reviewer_PZcm · 2024-10-29

**Soundness:** 2
**Presentation:** 2
**Contribution:** 2
**Rating:** 5
**Confidence:** 4

**Summary:**

This paper presents novel binarized convolutional neural networks (BCNNs), termed QB-Net and QSB-Net, which achieve state-of-the-art performance among BCNNs and are tested on edge devices. Experiments demonstrate the real-world deployment potential of QB-Net and QSB-Net.

**Strengths:**

1.The deployments on RPi 4B and Exynos strengthen the practical relevance of this work.
2.The results across various datasets and tasks validate the effectiveness of QB-Net.

**Weaknesses:**

1. The channel quadrupling technique is not novel; previous works, such as TResNet, have employed it. However, this work lacks a discussion on its application.

2. Figure 1(b) is unclear; moving the legend outside the figure may improve clarity.

3. The choice of model design requires a more thorough discussion.

4. The writing could be improved for clarity.

5. Additional tables in the Ablation Study section would be beneficial.

6. Could you clarify the impact of downsampling width first versus height first?

**Questions:**

see weaknesses

---

> ### Author Response · Authors · 2024-11-20
> **1st Rebuttal for Reviewer PZcm (1)**
>
> Dear Reviewer PZcm,
> Thanks for your valuable comments and idea. Considering your comments and ideas, we have prepared rebuttals, which address weak points and answer your question as follows:
>
> ---
>
>  * **Q1:** Novelty of the channel quadrupling technique.
>  * **A1:**  Thank you for your thoughtful comments. In line with your concerns, several existing works have adopted customized channel expansion during downsampling instead of simple channel doubling, as shown in the second paragraph of 1. Introduction section. However, the main contribution of our work is **not merely** proposing a structure for channel quadrupling. By applying the rule of channel quadrupling in the proposed models increases the number of channels in deeper layers, where we believe that it can lead to smoother entropy and a wider dynamic range in terms of frequency. The above explanation was written in the second paragraph of 1. Introduction section as:
> &nbsp;&nbsp; &nbsp;&nbsp;  ***"Our hypothesis suggests that the performance of BCNNs can be significantly influenced by the complexity of deep blocks and the configuration of downsampling. Although increasing the number of channels of all blocks by the same ratio can mitigate the issue, computational costs dramatically increase. Therefore, we think that developing a novel strategy for channel expansion and downsampling has the potential to achieve significant benefits in BCNNs."***
> In the revised version, the references with customized channel expansion TResNet and ResNet-50 will be added.
> &nbsp;&nbsp; &nbsp;&nbsp; Additionally, we have discussed that the channel quadrupling in QB-Net does not reduce the computational complexity of spatialwise convolutions with $stride=2$ in the downsampling blocks, as illustrated in Figures 1 (a) and 4 (a). Specifically, it does not decrease the operations (OPs) of spatialwise convolutions in deeper layers. We hypothesized that the information loss during downsampling is critical in binarized convolutional neural networks (BCNNs). Therefore, we expect that maintaining the OPs by quadrupling the number of channels during downsampling could help enhance the representation capacity.
> &nbsp;&nbsp; &nbsp;&nbsp; Furthermore, the proposed model also considers the reduction of computational costs. As described in Section 4.1, our motivation was to ensure that structural improvements in BCNNs primarily increase the model complexity in efficient components that significantly contribute to performance. By reducing the complexity of shallow layers in the proposed structure, the increased computational requirements in deeper layers can be mitigated, which distinguishes our approach from existing BCNNs.
> &nbsp;&nbsp; &nbsp;&nbsp;  Thank you again for your valuable feedback.
> ---
>
>  * **Q2:** Figure 1(b) is unclear; moving the legend outside the figure may improve clarity.
>  *   **A2:**  Thanks for your recommendation. According to your idea, the legends of the proposed models have been outside from Figure 1 (b) in the revised version.
>
> ---
>
>  * **Q3:**  Discussion of the choice of model design.
>  *   **A3:**  Thanks for valuable feedback. We have rewritten the expressions about the reason why the model structure was chosen. Now, the submitted previous version considered the benefits of using conventional depthwise separable convolutions and channel quadrupling as:
> &nbsp;&nbsp; &nbsp;&nbsp;  **Firstly, the channel quadrupling of QB-Net does not decrease the computational complexity for spatialwise convolution** with $stride=2$ in the downsampling blocks, as shown in **Figure 1 (a) and Figure 4 (a).** It does not decrease the OPs of spatialwise convolutions in deep layers. We hypothesize that the information loss during downsampling is critical in binarized convolutional neural networks (BCNNs), so that we expect that the maintained OPs by quadrupling the number of channels during downsampling can be helpful to provide more representation capacity. The smooth downsampling can provide additional representation capacity during downsampling, as shown in Figure 4 (a).
> &nbsp;&nbsp; &nbsp;&nbsp;  **Secondly,** by increasing channels in the deep layers with this proposed structure, we enhanced the representation capacity, which we showed both experimentally and logically to result in **smooth entropy** and **a wider dynamic range in terms of frequency.** Additionally, by considering the effect of the number of channels in the shallow and deep layers on performance, we derived a more lightweight structures of BCNNs.
> &nbsp;&nbsp; &nbsp;&nbsp;  **Finally,** for the detail discussion for effects of channel expansion and computational complexity in shallow and deep layers and smooth downsampling, **Appendix A.5 Detail Discussion for Ablation Studies have been added in the modified draft.**
> &nbsp;&nbsp; &nbsp;&nbsp;  We hope that the reviewers understand the hypotheses behind our model design and the structural contributions built upon them.

---

> ### Author Response · Authors · 2024-11-20
> **1st Rebuttal for Reviewer PZcm (2)**
>
> * **Q4:**  The writing for clarity.
>  *   **A4:**  We are sorry for the several uncomfortable points. In the revised version, several typos and misunderstandable expression have been revised.
> &nbsp;&nbsp; &nbsp;&nbsp;- "signficant" in line 85 has been corrected as "significant."
> &nbsp;&nbsp; &nbsp;&nbsp;-  teacher-studeint in line 215 has been corrected as "teacher-student."
> Besides, several awkward expressions have been revised. Thanks for your advice.
>
> ---
>  * **Q5:**   Additional tables in the Ablation Study section.
>  *   **A5:**  In agreement with your opinion and for the detail discussion for **effects of channel expansion and computational complexity in shallow and deep layers and smooth downsampling, Appendix A.5 Detail Discussion for Ablation Studies have been added in the modified draft.**, where **Table 7** has been added to summarize the effects of ablations and different model structures.
>
> ---
>  * **Q6:** Impact of downsampling width first versus height first.
>  *   **A6:**  Thanks for your good points. In the preliminary studies of this paper, we experimented with reversing the order of downsampling in the width and height directions during smooth downsampling. However, the difference was found to be marginal, with an impact of less than 0.2% on image classification performance on ImageNet-1K. Consequently, we propose a structure that performs downsampling in the width direction first, followed by downsampling in the height direction. Logically, unless the dataset has a specific bias, the order of downsampling is not considered critical. Based on this, we will provide clarification in the main text regarding the points you mentioned. Thanks again.
>
> ---
> Thanks all. Your valuable comments are helpful for enhancing the quality of this paper.

---

> > ### Comment · Reviewer_6HCY · 2024-11-26
> >
> > Some of my problems have been resolved, but I think the contribution of this paper is still limited.
> > Thus, I will change my rating from 5 to 6, but I could not give a higher rating.

---

> ### Author Response · Authors · 2024-11-24
> **Reminder from Authors regarding the Discussion Period**
>
> Dear Reviewer,
>
> We would like to kindly remind you that we submitted our responses to your comments last Wednesday. As the discussion period is approaching its end, we sincerely ask for your attention to review our response. If you find that our answers sufficiently address your concerns and questions, we would be grateful if you could consider revising your rating.
>
> Thank you very much for your time and effort, and we truly appreciate your valuable feedback.

---

> ### Author Response · Authors · 2024-11-29
> **Requesting for the feedback of the first rebuttal**
>
> Dear Reviewer PZcm,
>
> We would like to kindly remind you that we submitted our responses to your comments 9 days ago. As the discussion period is nearing its end, we would sincerely appreciate it if you could review our responses at your earliest convenience. If you find that our answers adequately address your concerns and questions, we would be grateful if you could consider revising your rating.
>
> Thank you very much for your time and effort. We look forward to receiving your valuable feedback.

---

> > ### Comment · Reviewer_PZcm · 2024-12-01
> >
> > Thank you for the author's response. After reviewing the rebuttal and the comments from other reviewers, I have decided on my final score of 5. My main concern remains the novelty of the work. It seems that, in terms of novelty, the contribution does not fully convince me, as it appears to offer somewhat incremental gains.

---

> ### Author Response · Authors · 2024-12-01
> **Answer for the Second Comments from Reviewer PZcm**
>
> Dear Reviewer PZcm,
>
> Thank you for your positive response and for considering our rating favorably.
> Regarding the concerns about the novelty of our work, we would like to present the following answeer:
>
> Regarding the novelty aspect, you raised concerns about quadruply expanding the channels . As we mentioned in our initial rebuttal and previous versions, such an approach can be found in FP32 models by custom-adjusting the width multiplier or through Neural Architecture Search (NAS). However, in the context of binary models, we have conducted **various analyses and empirical proof for our hypothesis to understand why this approach can lead to superior performance** in tasks like image classification, semantic segmentation, and object detection, while also reducing computational costs. This includes **analyses in the frequency domain and entropy characteristics.**  Notably, we have thoroughly validated  **the practical applicability of our binarized models on real hardware.** We hope that reviewers will consider these points positively.
>
> Once again, we appreciate your response. **If you and other reviewers have any further concerns during the remaining discussion period, please let us know, and we will do our best to address them.** Thank you again.

---

### Official Review · Reviewer_6HCY · 2024-11-01

**Soundness:** 3
**Presentation:** 3
**Contribution:** 3
**Rating:** 6
**Confidence:** 4

**Summary:**

This paper proposes novel binarized convolutional neural networks, which quadruple the number of channels and incorporate smooth downsampling.
It is a mixed model with FP32 depthwise separable and binarized pointwise convolutions.
It also starts with a few channels in shallow layers and expands them during downsampling by a factor of four, to improve the performance while reducing the total computational costs.
Besides, a novel smooth downsampling, channel-wise self-attention and multiple binarized convolutions in the fully connected layer are also used.
Experimental results demonstrate the efficiency of the proposed models in terms of performance, computational costs, and inference latency on real hardware on both ImageNet-1K classification and PASCAL VOC semantic segmentation.

**Strengths:**

1. The motivation is reasonable. They hypothesize that information loss during downsampling can impact model performance, and develop a novel strategy for configuring channel expansion and downsampling in BCNNs.
   2. The result is comparable with the state-of-the-art.

**Weaknesses:**

1. Typo: page 2 line 85, "signficant" should be "significant".
   2. Previous work [1] has proved the effectiveness of SE and its variant in BNNs, thus, it could not be regarded as a contribution.
   3. The author claims they mainly focus on performance, computational costs, and inference latency. However, multiplying binarized convolutions in the fully-connected layer aims to reduce storage costs, and has no relation to this claim.
   4. Why does the author quadruple the channel number? Could we set another expanding ratio like 3 or 5, instead of 4 used in this paper?

   [1] Training binary neural networks with real-to-binary convolutions. ICLR 2020

**Questions:**

As listed in weaknesses.

---

> ### Author Response · Authors · 2024-11-20
> **1st Rebuttal for Reviewer 6HCY**
>
> Dear Reviewer 6HCY,
> Thanks for your valuable comments and idea. Considering your comments and ideas, we have prepared rebuttals, which address weak points and answer your question as follows:
>
> ---
>
>  * **Q1:** Typo: page 2 line 85, "signficant" should be "significant".
>
>  * **A1:**  Thanks for finding the typo. We will revise it. Thanks again.
> ---
>
>  * **Q2:**  Contribution of self-attention.
>  *   **A2:**  Thanks for your good point. In agreement with your opinion, the self-attention (SE) has been used in several BCNNs. We do not think that the deployment of SE block is the main contribution. Instead, **we think that the applicability of SE blocks for achieving better performance should be analyzed because other existing BCNNs adopted SE blocks.** Therefore, in Abstract and Introduction of our draft, we clarify that self-attention is the **technique** for achieving better models.  Moreover, we emphasize that the SE block has proven to be **effective** and **useful**. We have deliberately mentioned a title as **applicability** of SE blocks in an independent subsection, **4.5 Applicability of Techniques for Better Models**, to mean that **it is not the main structural contribution of our work**.
> &nbsp;&nbsp; &nbsp;&nbsp; Our evaluation showed that the increase in latency was small while yielding considerable performance enhancements when SE blocks are deployed only after the DS convolutions during downsampling.
> &nbsp;&nbsp; &nbsp;&nbsp;We kindly ask for your understanding regarding this distinction.
>
> ---
>
>  * **Q3:**  Explanation of computational costs and inference latency for the fully-connected layer.
>  *   **A3:**  Thank you for your insightful questions. We have concluded that the computational reduction in the fully-connected (FC) layer in the proposed models is likely negligible in real hardware. Due to space constraints, we did not go into detail on this aspect in the main manuscript.
> &nbsp;&nbsp; &nbsp;&nbsp;In our models, where the number of classes for ImageNet-1K is 1000 and the number of channels in the final convolutional layer is 2048, the multiply-accumulate operations (MACCs) for the FC layer amount to approximately **2 million**. When $OPs = FLOPs + \frac{BOPs}{64}$, the $OPs$ for the proposed models range from **43 million to 69 million**, as shown in Table 3.
> &nbsp;&nbsp; &nbsp;&nbsp;While binarizing the FC layer could potentially reduce latency, **the latency contribution from the FC layer is relatively minor**. Therefore, we expect any speedup from binarizing the FC layer to be minimal. However, as shown in Table 3, binarizing the FC layer can significantly reduce storage costs, which can help mitigate the weakness of the increased number of channels in the FC layer. Due to these considerations, we limited our detail discussion on computational cost reduction from the binarized FC layer.
>
> ---
>  * **Q4:** Reason why expanding ratio is 4, not 3 or 5.
>  *   **A4:** Thanks for your sharp comments. In agreement with your concerns, the increasing number of channels in deep layer could be helpful to increase the model performance. The reason why we adopted the number **four** in the channel expansion during downsample is as follows:
> &nbsp;&nbsp; &nbsp;&nbsp;**Firstly, the channel quadrupling of QB-Net does not decrease the computational complexity for spatialwise convolution** with $stride=2$ in the downsampling blocks, as shown in **Figure 1 (a) and Figure 4 (a).** It does not decrease the OPs of spatialwise convolutions in deep layers. We hypothesize that the information loss during downsampling is critical in binarized convolutional neural networks (BCNNs), so that we expect that the maintained OPs by quadrupling the number of channels during downsampling can be helpful to provide more representation capacity.
> &nbsp;&nbsp; &nbsp;&nbsp;**Secondly, the structural benefits for implementing BCNNs were considered when quadrupling the number of channels**. In implementation of binarized operations, the number of bits in a word (64 bits) can be considered. In the optimization guide of Larq on real hardware [1], it was known that the number of channels should be expanded **with a factor of 8.** When quadrupling the number of channels during downsampling, the optimization guideline can be met. Thanks again for your good points.
>  &nbsp;&nbsp; &nbsp;&nbsp;[1] Tom Bannink, Adam Hillier, Lukas Geiger, Tim de Bruin, Leon Overweel, Jelmer Neeven, and Koen Helwegen. Larq compute engine: Design, benchmark and deploy state-of-the-art binarized neural networks. Proceedings of Machine Learning and Systems, 3:680?695, 2021.
>
> ---
> Thanks all. Your valuable comments are helpful for enhancing the quality of this paper.

---

> ### Author Response · Authors · 2024-11-24
> **Reminder from Authors regarding the Discussion Period**
>
> Dear Reviewer,
>
> We would like to kindly remind you that we submitted our responses to your comments last Wednesday. As the discussion period is approaching its end, we sincerely ask for your attention to review our response. If you find that our answers sufficiently address your concerns and questions, we would be grateful if you could consider revising your rating.
>
> Thank you very much for your time and effort, and we truly appreciate your valuable feedback.

---

> ### Author Response · Authors · 2024-11-26
> **Answer for Official Comment by Reviewer 6HCY**
>
> Dear Reviewer 6HCY,
>
> Thank you for your positive feedback on our rebuttal, which reflects the effort to address your concerns. We are especially grateful that you feel our response adequately addresses many of the concerns you raised. However, we would greatly appreciate it if you could provide more detailed feedback regarding any issues or questions related to the contribution. This would enable us to provide a clearer and more comprehensive explanation.
>
> We understand that all reviewers are very busy at this time, but since there is still some time left in the discussion period, we encourage you to share any additional questions or concerns you may have. Your insights would be incredibly valuable for improving our work.
>
> Once again, thank you for your encouraging response, and we look forward to hearing more from you.

---

### Official Review · Reviewer_KyRR · 2024-11-07

**Soundness:** 3
**Presentation:** 4
**Contribution:** 3
**Rating:** 6
**Confidence:** 4

**Summary:**

This paper introduces QB-Net and QSB-Net, novel binarized convolutional neural network architectures specifically designed for mobile environments. It designed "channel quadrupling" that starts with fewer channels in shallow layers and expands them four-fold during downsampling, along with a "smooth downsampling" technique.

**Strengths:**

1. Provides a fresh approach to BCNN design, moving beyond traditional methods.

2. The authors conduct real hardware testing on both Raspberry Pi 4B and Samsung Exynos processor, and the results demonstrate real-world applicability for mobile/edge devices. This is good.

**Weaknesses:**

1. A major concern is that the performance improvement is marginal or null when compared with SoTA BNN. This method performs worse than PokeBNN [1], which was published three years ago. PokeBNN has a top1 acc of 70.5 on ImageNet, while it has fewer flops.

2. Lacks deeper analysis of the relationship between model architecture and performance gains. I noticed that there are some performance degradation when trained from scratch. I am not sure if this performance gain from the designed modules themselves or from the training.

[1] Pokebnn: A binary pursuit of lightweight accuracy, CVPR 2022

**Questions:**

The proposed architectures combine FP32 depthwise separable convolutions with binarized pointwise convolutions, effectively reducing computational costs while maintaining performance through increased model complexity in deeper layers. It may improve the efficiency in terms of flops. **In system level, does the mix-precision design hurt the inference efficiency?**

---

> ### Author Response · Authors · 2024-11-20
> **1st Rebuttal for Reviewer KyRR (1)**
>
> Dear Reviewer KyRR,
> Thanks for your valuable comments and idea. Considering your comments and ideas, we have prepared rebuttals, which address weak points and answer your question as follows:
>
> ---
>
>  * **Q1:** Comparison with PokeBNN.
>  * **A1:**  Thanks for your good points. However, PokeBNN adopted **ResNet-50** as its backbone and teacher during its **long** training, so there are limitations when comparing it with the proposed model using ResNet-34 as its teacher. In addition, PokeBNN consists of a specific PokeInit block as it the first layer containing 8-bit operations and a PokeConv block containing 4-bit operations in its self-attention, which means that all layers of the PokeBNN contain 4-bit integer operations rather than binary operations. Besides, the outstanding performance of PokeBNN was based on the hyperparameter for the clipping bound. The above explanation was written in lines 133-137 of Related Works section as:
> &nbsp;&nbsp; &nbsp;&nbsp;***"Whereas PokeBNN (Zhang et al., 2022) achieved impressive results by adopting ResNet50 as its baseline and teacher models, its hyperparameter for the clipping bound and stride configuration in the first convolutional layer introduced a different optimization strategy. Different from Zhang et al. (2022), the proposed QSB-Net shows a novel strategy for channel quadrupling and smooth downsampling. Therefore, we conclude that the fundamental techniques and contributions of the proposed models are totally different from those of PokeBNN."***
> &nbsp;&nbsp; &nbsp;&nbsp; We considered it as a mixed-precision quantization model, not a BCNN model. Unlike the PokeBNN, the initial version of proposed models can be implemented using conventional FP32 and binarized operations. Therefore, we did not compare the proposed model with PokeBNN as our baseline in detail.
>
> ---
>  * **Q2:** Analysis of model architecture and performance gains and analysis for knowing performance gain from training or model structure.
>  *   **A2-1:**  **Analysis of relationship between model architecture and performance:** Thanks for your sharp points. We are sorry that I feel the model architecture optimization hasn't been sufficiently analyzed. However, we think that our hypothesis and structural analysis were considered in the manuscript as:
> &nbsp;&nbsp; &nbsp;&nbsp; **Firstly**, based on the hypothesis that the high representation capacity during downsampling in deep layer can be more beneficial, different structured models (Large and Small models of QB-Net and QSB-Net, and QSB-Net-SE) were analyzed, where **the block structures with different position of downsampling in deep layers and the existence of smooth downsampling** were given to know the detail model architectures and applicability of self-attention in BCNNs. As shown in Table 1 and Table 5, the different model structures based on our hypothesis were given. **For the detail discussion for effects of channel expansion and computational complexity in shallow and deep layers and smooth downsampling, Appendix A.5 Detail Discussion for Ablation Studies have been added in the modified draft.**
> &nbsp;&nbsp; &nbsp;&nbsp;**Secondly**, in Ablation Studies section, the structure of shallow layers with large feature maps, the effects of the number of channels in deep layers, learnable bias block, the applicability of 8-bit quantization in depthwise separable convolution and FC layers were analyzed.
> &nbsp;&nbsp; &nbsp;&nbsp;**Finally,** we hypothesized that the information loss in the downsampling is critical in BCNNs and demonstrated that these could be resolved through the proposed channel quadrupling and downsampling techniques. Therefore, by increasing channels in the deep layers with this proposed model structure, we enhanced the representation capacity, which we showed both experimentally and logically to result in **smooth entropy** and **a wider dynamic range in terms of frequency**.
> &nbsp;&nbsp; &nbsp;&nbsp; We think that the above analysis can show the relationship between the proposed model architectures and performance.

---

> ### Author Response · Authors · 2024-11-20
> **1st Rebuttal for Reviewer KyRR (2)**
>
> *   **A2-2: Performance gain from training:** Thanks for your good points. In agreement with your concerns, it is important to know the main cause of the enhanced performance gain. Because the baseline of proposed models was MobileNetV1, the training recipe of ReActNetA was adopted in 5.1 Section as:
> ***"We experimented with the proposed models on ImageNet-1K. Like ReActNetA (Liu et al.,2020), experiments adopted the two-stage teacher-student training using pretrained ResNet34 as a teacher (Hinton et al., 2015). In the first stage with 256 epochs, whereas the input features for BCONVs were binarized, weights were FP32 values. In the second stage, the pretrained weights from the first stage were used in the initialization. Both input features and weights for BCONVs were binarized during 256 training epochs. The detailed training process is described in Appendix A.4.**"*
> &nbsp;&nbsp; &nbsp;&nbsp; **By adopting the same training recipe**, the analysis shows that the proposed model structures can achieve high performance gain with reduced computational costs over the baseline.
> &nbsp;&nbsp; &nbsp;&nbsp; Besides, **the training from scratch in 5.3 Ablation Studies subsection** followed the training recipe of QuickNet, where the trained models significantly outperformed the counterparts.  In agreement with your opinion, performance can be enhanced using teacher-student training. However, compared with the results of counterparts such as ReActNetA and QuickNet based on the same training recipe, we assure that the proposed structure can be beneficial in increasing model performance.
>
> ---
>
>
>  * **Q3:**  In system level, does the mix-precision design hurt the inference efficiency?
>  *   **A3:** Thanks for good points. The mix-precision design in the proposed architecture, which combines FP32 depthwise separable convolutions with binarized pointwise convolutions, **achieves good inference efficiency at the system level even on general-purpose devices such as Raspberry Pi and Android phones, due to the use of inherent FP32 instructions, as shown Tables 3 and 4**. Notably, the computational cost of FP32 depthwise separable convolutions is lower than that of the 3x3 binarized convolutions used in ReActNet[1], further reducing the overall computational burden, **which was proved in reduced latency of the experiments on real hardware.** The following analysis can provide more clear proof as: Considering the OPs and latencies of ReActNetA as baseline, the ratios of OPs and latencies are compared in the below Table. Most cases denoted as bold have smaller ratios over those of OPs, which shows that the FP32 separable depthwise convolutions in the proposed models do not have negative impact on the inference efficiency.
>
> Table. Ratio of OPs and Latencies over baseline ReActNetA.
> | Model  | OPs Ratio | Latency RPi 4B Ratio | Latency Exynos Ratio |
> |:---:|:---:|:---:|:---:|
> | ReActNetA | 1 | 1  | 1 |
> | QB-Net-Small | 0.49| **0.46** | **0.47** |
> | QB-Net-Large | 0.61| **0.54** | **0.47** |
> | QSB-Net-Small | 0.61| 0.63 | **0.60** |
> | QSB-Net-Large | 0.71| 0.72 | **0.70** |
> | QSB-Net-Large(SE1) | 0.79| **0.75** | **0.73** |
>
> &nbsp;&nbsp; &nbsp;&nbsp; Of course, although the binarized depthwise separable convolution of MobiNet [2] could further reduce inference latency, the model performance can be significantly degraded, having only under 60% Top-1 accuracy on ImageNet-1K.
> &nbsp;&nbsp; &nbsp;&nbsp;While BCNNs leverage bit-level parallelism using standard instructions like XNOR and popcount on CPUs, enabling cost-effective AI inference without the need for GPUs or accelerators, the integration of efficient FP32 depthwise separable convolutions helps maintain performance while optimizing computational costs, compared with the baseline ReActNet. **Besides, the FP32 operations of depthwise separable convolutions and binarization operations such as XNOR and popcount in binarized pointwise convolutions were separated as a form of layers, so that FP32 and binarization operations are not mixed in the fine-grained levels.** The above combination in the proposed models ensures high inference efficiency and demonstrates strong practical performance on real system like Raspberry Pi and Android.
> &nbsp;&nbsp; &nbsp;&nbsp; [1] Zechun Liu, Zhiqiang Shen, Marios Savvides, and Kwang-Ting Cheng. Reactnet: Towards precise binary neural network with generalized activation functions. In European Conference onComputer Vision (ECCV), pp. 143?159, 2020.
> &nbsp;&nbsp; &nbsp;&nbsp; [2] Hai Phan, Yihui He, Marios Savvides, Zhiqiang Shen, et al. Mobinet: A mobile binary network for image classification. In The IEEE Winter Conference on Applications of Computer Vision, pp. 3453–3462, 2020a.
>
> ---
> Thanks all. Your valuable comments are helpful for enhancing the quality of this paper.

---

> ### Author Response · Authors · 2024-11-24
> **Reminder from Authors regarding the Discussion Period**
>
> Dear Reviewer,
>
> We would like to kindly remind you that we submitted our responses to your comments last Wednesday. As the discussion period is approaching its end, we sincerely ask for your attention to review our response. If you find that our answers sufficiently address your concerns and questions, we would be grateful if you could consider revising your rating.
>
> Thank you very much for your time and effort, and we truly appreciate your valuable feedback.

---

> ### Author Response · Authors · 2024-11-29
> **Requesting for the feedback of the first rebuttal**
>
> Dear Reviewer KyRR,
>
> We would like to kindly remind you that we submitted our responses to your comments 9 days ago. As the discussion period is nearing its end, we would sincerely appreciate it if you could review our responses at your earliest convenience. If you find that our answers adequately address your concerns and questions, we would be grateful if you could consider revising your rating.
>
> Thank you very much for your time and effort. We look forward to receiving your valuable feedback.

---

> > ### Comment · Reviewer_KyRR · 2024-12-02
> >
> > The comparison on RPi 4 is not an apples-to-apples comparison. The effectiveness (accuracy) should be maintained at the same level when comparing efficiency. In this context, ReActNetA is not an appropriate baseline. PokeBNN would be a more suitable comparison, and I think PokeBNN would demonstrate system efficiency improvement on RPi if compared with ReActNet. Therefore, I will maintain my score.

---

> ### Author Response · Authors · 2024-12-03
> **Answer About the First Answer of Reviewer KyRR**
>
> Dear Reviewer KyRR,
>
> As the discussion period is nearing its deadline, I’ve been eagerly awaiting your response and truly appreciate receiving it. Regarding the points you raised about PokeBNN’s computational efficiency on Raspberry Pi and the baseline comparison, I would like to address them as follows:
>
> ---
> Firstly, as described in the paper, **PokeBNN employs 8-bit computations (PokeInit) for initial convolution and 4-bit precision in self-attention layers.** Despite adopting **48 self-attention layers based on a ResNet-50 architecture,** it is reported to have a relatively low computational cost. However, we can counter this claim with the following points. Binary Convolutional Neural Networks (BCNNs) were initially develoed, as highlighted in XNOR-Net (2016), to leverage **bit-level parallelization on CPUs** that support only general floating or fixed-point and binary operations. For structures like PokeBNN to demonstrate their advantages, **8-bit and 4-bit parallel instructions must be supported.**
>
> Secondly, in our comparison based on the **Larq framework using Raspberry Pi 4B and Exynos, shown in Table 3,** it is not feasible to evaluate PokeBNN under such conditions. Moreover, whereas incorporating self-attention in all layers with a large number of such layers might reduce computational costs on PokeBNN, **in real hardware implementations, it can lead to increased latency, which in turn impacts power consumption.** In fact, whereas XNOR-Net introduces the calculation formula **$OPs = FLOPs + \frac{BOPs}{64}$, Table 3 in our study does not show such tremendous computational reductions.** This discrepancy arises because during inference, the sum of operations involving binary, 8-bit, and 4-bit computations exceeds the range of binary or mixed-precision operations, leading to **significant overhead from frequent conversions.**
>
> Therefore, **we believe it is essential to compare latency on real hardware and select an appropriate baseline for performance evaluation.** Assuming the availability of multi-level mixed-precision instructions, as seen in PokeBNN, and **comparing computational costs based solely on a table aggregation does not align with the goals of our research.**
>
> Different from the others, we proposed model structures aimed at addressing the hypothesis that the loss during downsampling plays a critical role in performance. **To clearly highlight the differences, we selected a baseline, explained the reason behind this choice, and sought experimental evidence to support it.** In this regard, our approach differs from previous works that merely presented existing architectures.
>
> ---
> **We hope this explanation clarifies our perspective and helps you and the reviewers better understand the approach we proposed.** We kindly request you to reconsider the rating in light of this discussion. Once again, thank you for your valuable feedback.

---

> > ### Comment · Reviewer_KyRR · 2024-12-03
> >
> > Thank you for your response. I got the idea of leveraging bit-level parallelization on CPUs that support only general floating or fixed-point and binary operations. I decide to raise my score.

---

> ### Author Response · Authors · 2024-12-03
> **Answer for the Second Response from Reviewer KyRR**
>
> Dear Reviewer KyRR,
>
> Thank you so much for carefully considering our explanations and for improving your rating. As the discussion period is coming to an end, we plan to send our final closing remarks to the reviewers tomorrow. We hope you will take our points into account in the future as well. Once again, thank you for your thoughtful feedback and support.

---

### Official Review · Reviewer_stXN · 2024-11-08

**Soundness:** 2
**Presentation:** 2
**Contribution:** 2
**Rating:** 5
**Confidence:** 4

**Summary:**

The paper introduces novel binarized convolutional neural networks (BCNNs), QB-Net and QSB-Net, designed to enhance performance in low-cost mobile environments by quadrupling the number of channels and incorporating smooth downsampling. The models combine FP32 depthwise separable (DS) convolutions with binarized 1×1 pointwise convolutions to reduce computational costs. The proposed structure starts with a small number of channels in shallow layers and expands them during downsampling, managing model complexity effectively. Experimental results demonstrate improved performance.

**Strengths:**

1. The proposed QB-Net and QSB-Net models offer a new approach to channel expansion and downsampling in BCNNs, which is a  contribution to the field of model compression and quantization.
2. The paper demonstrates substantial performance improvements over existing BCNNs, which is a strong contribution for mobile and edge computing applications.
3. The paper provides a detailed explanation of the technical methods, including the new structure for channel quadrupling and smooth downsampling techniques.

**Weaknesses:**

1. The paper may be seen as more of an engineering solution rather than a contribution to the foundational knowledge of the field.
2.  Although the paper claims reduced computational costs, the quadrupling of channels and the introduction of smooth downsampling could potentially increase the model's complexity, especially in deeper layers. This could lead to higher computational requirements that may offset the benefits of reduced pointwise convolution costs, especially the additional connections..
3. The abstract mentions improvements in inference latency on real hardware, but it does not discuss the energy efficiency of the models. For mobile environments, energy consumption is a critical factor, and the proposed models' impact on this aspect should be evaluated.

**Questions:**

Mixed use of fp32 and binary value will increase the complexity of hardware design, how to solve it?

---

> ### Author Response · Authors · 2024-11-20
> **1st Rebuttal for Reviewer stXN**
>
> Dear Reviewer stXN,
> Thanks for your valuable comments and idea. Considering your comments and ideas, we have prepared rebuttals, which address weak points and answer your question as follows:
>
> ---
>
>  * **Q1:** Concerns of an engineering solution rather than a contribution to the foundational knowledge of the field.
>  * **A1:**  Thank you for your feedback. While certain structural decisions were influenced by experimental research, **we hypothesized that the information loss in the downsampling is critical in BCNNs and demonstrated that these could be resolved through the proposed channel quadrupling and downsampling techniques.** By increasing channels in the deep layers with this proposed structure, we enhanced the representation capacity, which we showed both experimentally and logically to result in **smooth entropy** and **a wider dynamic range in terms of frequency.** Additionally, by considering the effect of the number of channels in the shallow and deep layers on performance, we derived a more lightweight structures of BCNNs, which we believe **contributes to foundational knowledge improvements in BCNNs.**
> ---
>
>  * **Q2:**  Concerns about increase in the model's complexity, especially in deeper layers.
>  *   **A2:**  Thanks for your sharp points. In agreement with your concerns, the computational complexity can be increased in deep layer by quadrupling the number of channels during downsampling. **However, as shown in 4.1 Motivations section, our motivation was that the structural development of BCNNs should mainly increase the model complexity of the efficient part that significantly contributes to model performance.** When reducing the complexity of shallow layers in the proposed structure, the increasing computational requirements in deep layer can be mitigated.
> &nbsp;&nbsp; &nbsp;&nbsp;   In conclusion, the efficient part to the performance of BCNNs can be the deep layer in our findings. While computational complexity of the deep layers can increase, the decreasing complexity of the inefficient part (shallow layers) can reduce the total inference latency, which can be naturally achieved by quadrupling channels during downsampling.
> ---
>
>  * **Q3:**  Discussion about the energy efficiency of the models.
>  *   **A3:**  Thanks for your good points. We have added A.11 Estimation of Energy Consumption on Real Hardware in the revised draft. In the evaluation, we measured the current drawn from the USB power line using a current probe over 500 test runs. When the models were not being evaluated, the current consumption was approximately 713 mA @ 5V. In contrast, during the execution of the models listed in Table 5, the current consumption ranged from 912 mA to 936 mA. Considering the resolution of the current probe, we thought that these variations were not significant. **Based on these observations, we conclude that the systematic energy consumption is largely proportional to inference latency because no substantial differences in system power consumption were detected during the evaluations.**
> &nbsp;&nbsp; &nbsp;&nbsp; Several existing works based on FPGA implementation and accelerators can estimate **the detail power consumption from the hardware synthesis tool.** However, because we adopted real hardware system, simple hardware power profiling can be the best choice to estimate energy consumption.
> &nbsp;&nbsp; &nbsp;&nbsp; We hope that the reviewers understand that the methods for predicting energy consumption related to FPGA and accelerator silicon implementations must inherently differ from those for predicting energy consumption based on CPU instructions on real edge devices.
>
>
> ---
>  * **Q4:**  Solution of mixed use of fp32 and binary value that increases the complexity of hardware design.
>  *   **A4:**  Binarized convolutional neural networks (BCNNs) leverage bit-level parallelism inherent in standard CPUs, as mentioned in the first section of the Introduction section. This approach enables parallel processing using general-purpose instructions like XNOR logical operations and population count (popcount) without relying on GPUs or accelerators, making it suitable for devices such as Raspberry Pi and Android platforms. Unlike 8-bit, 4-bit quantization or specialized 8-bit or 16-bit floating-point AI-aware instructions, **the proposed BCNNs can utilize standard instructions,** facilitating the implementation of cost-effective and high-speed AI inference **without the need for dedicated hardware design.**
>
> ---
> Thanks all. Your valuable comments are helpful for enhancing the quality of this paper.

---

> > ### Comment · Reviewer_stXN · 2024-11-25
> >
> > Thank you for your detailed rebuttal. I appreciate the time and effort you have put into addressing the concerns raised in the initial review. The design of QB-net and QSB-net is  an engineering solution and the application is still unclear. I maintain my rate.

---

> > > ### Author Response · Authors · 2024-11-25
> > > **First Answer for Official Comment by Reviewer stXN**
> > >
> > > Thank you so much for your valuable feedback and for your positive evaluation of our rebuttal.
> > >
> > > Regarding the engineering solution, we would like to share our thought as:
> > >
> > > Binarized convolutional neural networks (BCNNs) inherently require some trade-offs in performance to achieve lightweight models. Therefore, it is inevitable to develop models while considering factors such as latency and computational costs. We are sorry if this aspect is perceived as the engineering solution. However, we believe that the structure of our model, including rules like channel quadrupling to increase the complexity of deep layers, is supported by a theoretical background and evidence.
> > >
> > > As for the term **application,** we are not entirely sure what is being referred to, but we would like to emphasize that binarized models have broad application potential. Even in many embedded systems without high-performance accelerators like GPUs or NPUs, they can be implemented through basic operations such as XNOR computations and 1-bit counting.
> > >
> > > Although there is limited time remaining in the discussion period, we would greatly appreciate it if you could consider these aspects. Thank you once again for your kind feedback.

---

> ### Author Response · Authors · 2024-11-24
> **Reminder from Authors regarding the Discussion Period**
>
> Dear Reviewer,
>
> We would like to kindly remind you that we submitted our responses to your comments last Wednesday. As the discussion period is approaching its end, we sincerely ask for your attention to review our response. If you find that our answers sufficiently address your concerns and questions, we would be grateful if you could consider revising your rating.
>
> Thank you very much for your time and effort, and we truly appreciate your valuable feedback.

---

### Author Response · Authors · 2024-11-22
**Request for Review of the Rebuttal**

Dear Reviewers,

We have prepared this rebuttal by addressing your comments from two days ago. We understand you have many commitments and a busy schedule, but we hope to discuss our responses to your comments. If our rebuttal sufficiently addresses your concerns, we kindly ask you to reconsider your ratings.

Thank you.

---

### Author Response · Authors · 2024-11-27
**Gentle Reminder for deadline of PDF modification**

Dear Reviewers,

As the deadline for finalizing the revised version of the paper's PDF approaches, further modifications to the PDF are no longer possible. If reviewers raise additional discussions or concerns, we will do our best to address them thoroughly in the rebuttal. To respond to the concerns raised by the five reviewers, we have added several appendices and made revisions and additions to the figures and tables. Please review these sections, and **if you find that the concerns you raised have been adequately addressed, we kindly ask you to reconsider your rating.** Once again, we sincerely thank the reviewers for their valuable feedback.

---

> ### Comment · Reviewer_d56q · 2024-11-28
> **RE: Gentle Reminder for deadline of PDF modification**
>
> Dear authors,
>
> I would like to express my sincere appreciation for your detailed feedback and the additional experiments you have provided. I concur that the current approach represents a robust solution from an industry application perspective.
>
> However, as this work is being considered for a top-tier machine learning conference, a significant concern arises regarding its potential to offer deeper insights into binary networks or to inspire future research. While the approach is commendable, it may be more aligned with the focus of venues like MLsys, which emphasize deployment. I would consider this paper to be above the baseline if it were to be presented at MLsys, given its strong emphasis on practical deployment.
>
> Thank you once again for your thoughtful contributions.
>
> Best regards

---

### Author Response · Authors · 2024-11-28
**Answer for Concerns from Reviewer d56q**

Dear Reviewer d56q,

Thank you so much for reading our paper and providing various feedback. Regarding the concerns raised in the paper, our approach is as follows.
Firstly, we would like to acknowledge that there have been numerous approaches to binary neural networks (BNNs) in the literature. Many prior works on BNNs have optimized performance through **specific hyperparameters** or **employed customized sign functions,** leading to many publications in top-tier conferences. Additionally, we would like to point out that many of these studies only focus on image classification tasks.

However, we believe that such approaches can often result in **excessively long training times** for a single epoch, or performance improvements that are merely coincidental due to **repeated hyperparameter tuning**, making them **less practical.**
Instead, we chose to focus on **structural changes to the model** and carefully examined whether **the design rules we applied could create an effective lightweight model.**

In particular, we explored the effects of adding more complexity during downsampling and analyzed the resulting changes in entropy and frequency domain characteristics. Through this, **we aimed to explain the phenomena observed and demonstrate that our model successfully achieves its fundamental goals of lightweight design and performance improvement.**

We believed this makes our work a candidate for a top-tier conference.

Once again, we sincerely appreciate your feedback on our paper. With six days remaining, we welcome any additional comments or suggestions, as they would greatly help us further improve the quality of our work. Thank you again, Reviewer d56q.

---

### Author Response · Authors · 2024-12-03
**Final Answer for Reviewers and ACs**

Dear Reviewers and ACs,



Thank you for your valuable feedback during the discussion period. Your insights have helped us enhance the quality of our revised paper and refine the clarity of various ideas. However, we feel that some of the concerns raised may stem from misunderstandings or a lack of clarity in explaining our main contributions. As such, we would like to take this opportunity to summarize the remaining concerns and provide detailed explanations of our revisions. **Most of all, we ask the reviewers to consider not just the similarity in components but also the reasoning behind our architectural choices and the evidence we provide to support them.**


----------


* Key Remaining Concerns


1. Lack of novelty and perceived incremental nature of the work. The research is seen as focusing on the model architecture itself rather than offering insightful contributions.

2. Lack of clarity in application scenarios.



----------



**1. Lack of Novelty and Perceived Incremental Nature of the Work**



**Response:**

We kindly request the reviewers to evaluate not only the similarity in components but also the rationale behind our architectural decisions, along with the supporting efforts we have presented. **Notably, our work is rooted in a clear and motivated hypothesis:** that information loss during downsampling is a fundamental bottleneck in BCNNs. To address this, we introduced **channel quadrupling** and **advanced downsampling techniques.** These are not arbitrary design decisions but are grounded in the premise that **increasing representation capacity in deeper layers preserves richer feature representations, with reduced computational costs.**



This approach significantly enhances the model's ability to achieve **smoother entropy** and **broader dynamic range in frequency,** a crucial performance improvement substantiated through rigorous analysis. **We hope the reviewers will recognize this theoretical motivation and the systematic steps we took to validate it.**



----------


**2. Lack of Clarity in Application Scenarios**



**Response:**

We believe that **BCNNs are well-suited for high-performance microcontroller-level AI models,** where CPU-based bit-level parallelization is advantageous. Our target applications include platforms such as Raspberry Pi 4B and mobile devices, and we validated the performance of our model in these scenarios. Besides, previous BCNNs have only been evaluated on the Pascal VOC benchmark which is a performance limitation. However, we are the first BCNN model to evaluate **semantic segmentation and object detection on the COCO-Stuff benchmark**. Therefore, we demonstrate the potential applicability of the proposed model with improved results compared to models with FP32 precision.



----------



We deeply appreciate the reviewers' feedback, though we wish the discussion had provided more specific concerns to address or areas to elaborate on. This would have allowed us to offer even more detailed clarifications during the discussion period.



We kindly request the reviewers to reconsider the ratings in light of the above explanations, and we sincerely thank you again for your thoughtful feedback.

---

### Meta-Review · Area_Chair_LPNa · 2024-12-22

**Metareview:**

The paper proposed a binarized convolutional neural network utilizing channel quadrupling and smooth downsampling, and presented reasonable results to demonstrate the contribution of the paper. The reviewers had some concerns on the following: first, they feel the novelty is insufficient although the motivation seems intuitive. Second, they argue that the performance improvements over the existing strong competitors are marginal, which thus cannot fully reflect the advantages of the proposal. Lastly, the reviewers are not happy about the presentation quality because of typos and unclarity in the submission. Based on these comments, AC decided to recommend a rejection for this time.

**Additional Comments On Reviewer Discussion:**

The reviewers requested clarification and additional results in the rebuttal. However, the rebuttal from the authors did not fully address the concerns of the reviewers.

---

### Decision · Program_Chairs · 2025-01-22

Reject